# Comparing black carbon and aerosol absorption measuring instruments – a new system using lab-generated soot coated with controlled amounts of secondary organic matter

Daniel M. Kalbermatter[1], Griša Močnik[2,3,4] Luka Drinovec[2,3,4], Bradley Visser[5], Jannis Röhrbein[5], Matthias Oscity[5], Ernest Weingartner[5], Antti-Pekka Hyvärinen[6] and Konstantina Vasilatou[1]

[1]Laboratory Particles and Aerosols, Federal Institute of Metrology METAS, Bern-Wabern, 3003, Switzerland
[2]Center for Atmospheric Research, University of Nova Gorica, Nova Gorica, 5270, Slovenia
[3]Haze Instruments d.o.o., Ljubljana, 1000, Slovenia
[4]Department of Condensed Matter Physics, Jozef Stefan Institute, Ljubljana, 1000, Slovenia
[5]Institute for Sensors and Electronics, University of Applied Sciences Northwestern Switzerland FHNW, Windisch, 5210, Switzerland
[6]Finnish Meteorological Institute, Helsinki, 00560, Finland

*Correspondence to*: Konstantina Vasilatou (konstantina.vasilatou@metas.ch)

**Abstract.** We report on an inter-comparison of black carbon and aerosol absorption measuring instruments with laboratory-generated soot particles coated with controlled amounts of secondary organic matter (SOM). The aerosol generation setup consisted of a miniCAST 5201 Type BC burner for the generation of soot particles and a new automated oxidation flow reactor based on the micro smog chamber (MSC) for the generation of SOM from the ozonolysis of α-pinene. A series of test aerosols was generated with elemental to total carbon (EC/TC) mass fraction ranging from about 90 % down to 10 % and single scattering albedo (SSA at 637 nm) from almost 0 to about 0.7. A dual-spot aethalometer AE33, a photoacoustic extinctiometer (PAX, 870 nm), a multi-angle absorption photometer (MAAP), a prototype photoacoustic instrument and two prototype photo-thermal interferometers (PTAAM-2λ and MSPTI) were exposed to the test aerosols in parallel. Significant deviations in the response of the instruments were observed depending on the amount of secondary organic coating. We believe that the setup and methodology described in this study can easily be standardized and provide a straightforward and reproducible procedure for the inter-comparison and characterisation of both filter-based and in situ BC-measuring instruments based on realistic test aerosols.

## 1 Introduction

Black carbon (BC)-containing particles are produced from incomplete combustion of fossil fuels or biomass. BC is believed to be the second most significant radiative forcing agent after carbon dioxide (Bond et al., 2013; Ramanathan and Carmichael,

2008). However, its influence on the radiative balance of the earth cannot be easily quantified because BC particles in ambient air are usually internally mixed with organic and/or inorganic species, which may cause absorption enhancement through the so-called "lensing effect" (Cappa et al., 2012; Liu et al., 2015).

Despite a plethora of commercially available BC-monitoring instruments based on different measurement techniques, quantification of BC mass concentration remains to this day a challenge. Deviations between 15 % and 30 % among instruments of the same type (Cuesta-Mosquera et al., 2021; Müller et al., 2011a) and up to 50–60 % for instruments of different measurement principle (Chirico et al., 2010; Slowik et al., 2007) have been reported. Among all commercial BC monitors, filter-based absorption photometers, such as the aethalometer, are the most widely used at air quality monitoring

stations thanks to their robust design. At the same time, these instruments are the most prone to measurement artefacts due to the use of filters for collecting the particles. Even though correction algorithms have been proposed for minimizing measurement biases (see (Collaud Coen et al., 2010) and references therein; (Drinovec et al., 2015) for a measurement of the loading bias), no satisfactory solution has been found for quantifying the absorption coefficient or for determining site-independent equivalent BC (eBC) mass concentrations.

To compare the performance of different instruments or to investigate unit-to-unit variability, several field and laboratory-based inter-comparisons of BC-monitoring instruments have been conducted in the past. Slowik et al. compared a single particle soot photometer (SP2), a multi-angle absorption photometer (MAAP), and photoacoustic spectrometer (PAS) with uncoated soot generated by a McKenna burner and soot coated with organic material, such as oleic acid and anthracene (Slowik et al., 2007). In another study, soot generated by a McKenna burner was coated with sulfuric acid and dioctyl sebacate (DOS),

and the effect of non-absorbing coatings on the response of filter-based and in situ BC-measuring instruments was determined (Cross et al., 2010). Holder et al. compared an SP2, a three-wavelength photoacoustic soot spectrometer (PASS-3) and an aethalometer (AE-42) during on-road and near-road measurements (Holder et al., 2014) while Tasoglou et al. compared six commercially available BC-measuring instruments using aerosols from biomass burning (Tasoglou et al., 2018). Moreover, two workshops with a large set of aerosol absorption photometers were conducted in 2005 and 2007, revealing a large variation

in the response to absorbing aerosol particles for different types of instruments (Müller et al., 2011a). More recently, an inter-comparison of 23 aethalometers was carried out with synthetic particles (soot generated by a miniCAST burner, nigrosin particles) and ambient air to investigate the individual performance of the instruments and their comparability (Cuesta-Mosquera et al., 2021).

Experiments in large-scale smog chambers are also conducted to simulate atmospheric ageing of soot particles and investigate

the response of the instruments to secondary organic coating (Cappa et al., 2008; Chirico et al., 2010; Weingartner et al., 2003). Whilst smog-chamber studies allow for controlled laboratory experiments with realistic test aerosols, they are time-consuming, with each measurement ranging up to a few days (Weingartner et al., 2003). Consequently, such experiments are typically

restricted to the generation of a single or a limited number of test aerosol types. A reliable inter-comparison of BC-measuring instruments with a series of different ambient-like aerosols would not be possible in a reasonable timeframe.

Recently, a compact and user-friendly setup based on a miniCAST combustion generator and an oxidation flow reactor (OFR) known as micro smog chamber (MSC) was proposed for the controlled generation of fresh and aged soot particles in the laboratory (Ess et al., 2021a). A series of test aerosols simulating a wide range of optical properties and elemental to total carbon (EC/TC) mass fraction could be generated within a few hours as opposed to a few days with conventional smog chambers. Compared to other OFRs reported in the literature (George et al., 2007; Kang et al., 2007), the MSC is designed to
operate at much higher aerosol loads which can subsequently be diluted, thus generating aerosols at high flow rates but still sufficiently high number concentrations to simultaneously feed multiple devices.

This study moves beyond the work by Ess et al. (Ess et al., 2021a) by demonstrating in practice how soot particles coated with controlled amounts of secondary organic matter (SOM) from the ozonolysis of α-pinene can be used to challenge a large number of BC-measuring instruments in parallel. More specifically, a dual-spot aethalometer, a photoacoustic extinctiometer
(PAX, 870 nm), a MAAP, a prototype photoacoustic instrument (PAS) and two prototype photo-thermal interferometers (PTAAM-2λ and MSPTI) were exposed to a series of aerosols with EC/TC mass fraction ranging from > 90 % down to 10 % and single scattering albedo (SSA) from almost 0 to about 0.7. The PTAAM-2λ is now commercially available (Haze Instruments, 2021) and this is the first time it has been compared to a range of established instruments. We believe that the setup and methodology described in this study can easily be standardized and provide a straightforward and reproducible
procedure for the inter-comparison and characterisation of both filter-based and in situ BC-measuring instruments based on realistic test aerosols.

## 2 Methods

### 2.1 Aerosol generation

Soot particles were generated by a miniCAST 5201 Type BC (Jing Ltd., Switzerland), hereafter referred to simply as
miniCAST BC, as previously described in Ess et al. (Ess et al., 2021b; Ess and Vasilatou, 2019). Two operation points in the "premixed flame mode" were used, both resulting in particles of roughly 90 nm mobility diameter (see Table S1 for gas flows if the operation points). The sample flow was dried using a diffusion dryer (Silicagel orange Perlen, Dry & Safe GmbH, Switzerland).

A novel "organic coating unit" (OCU, FHNW, Switzerland, (Keller et al., 2021a, 2021b)) was used to coat soot particles with
secondary organic matter. The process is described in (Ess et al., 2021a); however, the OCU combines an optional humidifier (not used in this study), a dosing system for up to two volatile organic compounds (VOC1 and VOC2, see Fig. 1) and an

oxidation flow reactor in an integrated unit. The soot was mixed in the OCU with α-pinene vapours (VOC1), which were held at constant concentration using the integrated dosing system and built-in photo-ionisation detector (PID-A1 Rev 2, Alphasense Ltd, UK). The PID sensor was regularly calibrated using a 100 ppm isobutylene–air mixture. The OCU was only used for the

coated operation points, for the uncoated operation points an identical setup without the OCU was used.

Two variations of the setup were used in this study. For "Setup 1" the soot aerosol generated by the miniCAST BC was delivered undiluted to the OCU while for "Setup 0.1" the aerosol was diluted at a 1:10 ratio with dry air (VKL 10 dilution unit, Palas GmbH, Germany) as shown in Fig. 1. The aerosol relative humidity before coating was about 25 % and <5 % for Setup 1 and 0.1, respectively.

After generation, the aerosol was further diluted by a rotational diluter (MD19, Matter Engineering AG, Switzerland) using different dilutions depending on the experiment. As the sample flow after the diluter (9.5 L min$^{-1}$) was not enough for all the instruments under test, an additional dilution stage was built using dry filtered air provided through an MFC and a mixing volume. The aerosol was then first split up to the high-volume instruments (MAAP and nephelometer) before using a second 19-port flow splitter for all the other instruments (see Fig. 1 for a schematic overview). The design of the custom-made flow

splitter is shown in Fig. S1 of the supplementary information. The splitter bias was determined as per (ISO, 2015) and found to be around 1 %. The PAX, PTAAM, PAS and AE33 have very similar sampling flows (1-2 L/min) and the difference in diffusion losses was compensated by adapting the length of the connecting tube to the flow of the instrument. For the MSPTI, which has a flow of 0.25 L/min, the connecting tube was kept as short as possible. Possible differences in the internal path length of the instruments (between the aerosol inlet and measurement cell) were not taken into account. In the case of the

MAAP, which was operated at a flow of 12 L/min, it was challenging to compensate for the difference in the diffusion losses. We cannot rule out that the measurements by the MAAP are biased because of lower diffusion losses in the connecting tube, but we estimate that this bias is <5 % and therefore much smaller than the systematic uncertainties of this filter-based instrument.

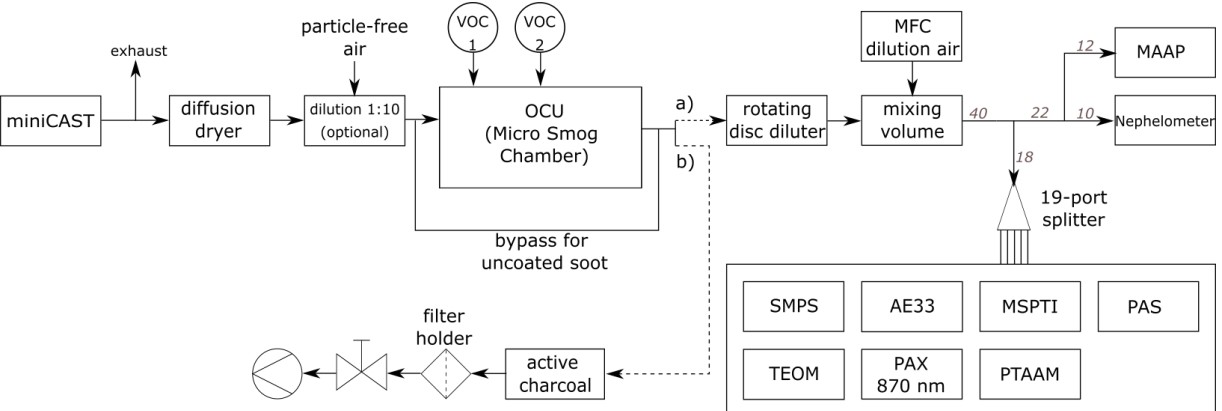

Figure 1. Schematic of experimental setup. The numbers above the arrows indicate aerosol flows in L min$^{-1}$.

## 2.2 BC- and aerosol-absorption-measuring instruments

The dual-spot aethalometer (AE33 aethalometer, Magee Scientific, Berkeley, USA) is a filter-based absorption photometer (Drinovec et al., 2015). It measures at seven different wavelengths (370–950 nm). To correct for filter-loading artifacts, the device measures the change of light attenuation at two distinct filter spots loaded at different flow rates. A standard multiple-scattering parameter $C$=1.39 (provided by the manufacturer) was applied to obtain the absorption coefficient from the measured attenuation coefficient. The aethalometer was operated at a sample flow of 2 L min$^{-1}$ and a temporal resolution of 1 min. Absorption Ångström exponents (*AAE*) were calculated using absorption coefficients measured by the aethalometer over all wavelengths (Drinovec et al., 2015). We estimate the measurement uncertainties of the AE33, based on our measurements and cross-sensitivity to scattering (Yus-Díez et al., 2021) to be around 20% (for a coverage factor $k$=1), consistent with the 25% value from (WMO, 2016).

The Thermo Scientific Model 5012 Multi-Angle Absorption Photometer, MAAP is a filter-based instrument that measures aerosol absorption at a nominal wavelength of 670 nm. The filter loading-related artifacts affecting the determination of absorption coefficient are taken into account in the design of the instrument. This is done by incorporating light transmittance and reflectance measurements at multiple angles and by implementing a radiative transfer calculation in the internal programming of the instrument. The absorption coefficient $b_{abs}$ for MAAP has been derived throughout the manuscript by

$$b_{abs} = M_{BC} \cdot Q_{BC} \cdot 1.05 \tag{1}$$

where $M_{BC}$ is the mass concentration of black carbon, $Q_{BC}$ is the specific absorption coefficient of 6.6 m$^2$g$^{-1}$ of MAAP, and 1.05 is a factor to correct the absorption coefficient to the true wavelength of the instrument light source, 637 nm (Müller et al., 2011a). According to (Petzold and Schönlinner, 2004), the measurement uncertainty in $b_{abs}$ is estimated to be 12%.

Periodic variations in the $b_{abs}$ measurements performed with the MAAP were observed especially at high black carbon concentrations during the campaign (see Fig. S2 for an example). While the MAAP is known to suffer from a measurement artefact occurring at high concentrations (Hyvärinen et al., 2013), the observed variations were unrelated to it. While the exact reason for the variations is not known, they occur mid-range of the MAAP spot collection duration, and thus seem to be instrument-dependent and possibly related to a non-disclosed internal averaging algorithm.

A photoacoustic extinctiometer PAX (PAX 870 nm, Droplet Measurement Technologies Inc., Boulder, Colorado, USA) was also used. The PAX measures absorption and extinction in parallel by combining a photoacoustic cell with an integrating nephelometer (Arnott et al., 1999). The PAX was operated at a sample flow of 1 L min$^{-1}$ and an averaging time of 1 min with a 1 min zero measurement every 5 min. The estimated uncertainties for the absorption measurements are ≤11 % according to (Nakayama et al., 2015).

The prototype photoacoustic sensor (PAS) from the FHNW group uses three different wavelengths (445 nm, 520 nm, 638 nm, ~300 mW each) for in-situ light absorption measurements. The diode-lasers are guided into a metallic resonator with elliptical cross-section along a focal point and are modulated at ultrasonic frequencies (~23.7 kHz) sequentially every minute for each wavelength. The modulation frequency adapts every 5 to 10 min (thermal drift) to match the resonance frequency of the resonator cell. The resulting standing wave is measured with a digital microphone placed in the middle of the resonator cell (long axis) at the other focal point of the ellipse. The signal is then preamplified and demodulated with a Stanford SR850 Lock-In Amplifier (Stanford Research Systems, Sunnyvale, California, USA). Section S4 describes the measurement principle, the device and the motivation behind it in more details. The instrument was calibrated using nitrogen dioxide (NO$_2$) and operated at 1 L min$^{-1}$. Due to technical issues during the campaign (discussed in Section 3), the uncertainty in the determination of the absorption coefficient can be high, reaching at times 100%. For this reason, the PAS measurements are presented in the Supplemental Information.

The photo-thermal interferometer PTAAM-2λ is based on a folded Mach-Zehnder interferometer design (similar to (Moosmüller and Arnott, 1996; Sedlacek, 2006; Visser et al., 2020)). The He–Ne probe laser beam is split into the sample chamber and reference beams. Pump lasers at 532 and 1064 nm are modulated at different frequencies and focused in the sample chamber using an axicon for concurrent measurement of the same sample. The quadrature point is maintained using a pressure cell. The interferometer signal is detected by two photodiodes and resolved by a dual-channel lock-in amplifier measuring at the two respective frequencies. The green channel is calibrated using NO$_2$. The calibration is transferred to the

infrared (IR) channel using aerosolized nigrosin and its relative green-to-infrared absorption ratio, determined using a Mie calculation based on size distribution measurements. The verification at 532 nm shows a 6 % difference between the Mie calculation and the calibrated measurements of the absorption coefficient. The combined measurement uncertainties (coverage factor $k$=1) for the absorption coefficients at 532 nm and 1064 nm, and the absorption Angstrom exponents (AAE) are 6% (532 nm), 8% (1064 nm), 9% (AAE) (Drinovec et al., 2020, 2021).

The photo-thermal interferometer MSPTI is an improved version of the instrument presented in (Visser et al., 2020). Briefly, the instrument design is similar to a folded Mach-Zehnder interferometer (Moosmüller and Arnott, 1996; Sedlacek, 2006), with the optical elements in the interferometer consisting of a combined beam splitter and mirror block and a retroreflector. In contrast to the PTAAM-2 and other photo-thermal interferometers, the MSPTI operates with only a single modulated laser (Nd:YAG, 532 nm), which is employed as both the pump and probe beam. This beam is split 50–50 and one of the resulting beams is sent through the sample chamber, whereas the other traverses the reference chamber. The beams are recombined at the beam splitter, resulting in interference patterns. In these experiments the filtered sample (HEPA grade absolute filter) is employed as the "zero" sample in the reference arm of the interferometer. Phase quadrature is maintained via an improved version of the pressure cell from (Visser et al., 2020). The MSPTI is calibrated using $NO_2$ and was operated at a flow rate of 0.25 L min$^{-1}$. The combined measurement uncertainty ($k$=1) for the absorption coefficient at 532 nm is estimated to be about 13%.

## 2.3 Additional aerosol characterization

Mobility size distribution and number concentration were measured using a scanning mobility particle sizer SMPS (Electrostatic Classifier Series 3080 with $^{85}$Kr radioactive source, DMA column 3081, CPC 3776 low flow, TSI Incorporated, USA). The DMA was operated with a sheath air of 3 L min$^{-1}$ and a sample flow of 0.3 L min$^{-1}$. Geometric mean mobility diameter ($GMD_{mob}$) and total number concentrations were determined from the size distribution using the software provided with the instrument (Aerosol Instrument Manager, v 9.0.0.0, TSI Incorporated, USA). The ratio of total number concentrations between the different operation points was used to scale $b_{abs}$ values reported by the BC-measuring instruments in order to account for day-to-day variability, with the concentration of the uncoated operation points being the reference.

An integrating nephelometer (AirPhoton model IN101) was used to measure light scattering coefficients over the angular range from 7 to 170 degrees. The AirPhoton IN101 utilizes LED light sources to make measurements at 450 nm, 532 nm and 632 nm. According to the manufacturer, the truncation correction for the AirPhoton should be done identically to TSI model 3563. However, as the single scattering albedo in the experiments was substantially below the specifications of the correction scheme proposed by the manufacturer (Müller et al., 2011b), no truncation correction was applied to the data.

A Tapered Element Oscillating Microbalance (TEOM 1405, Ambient Particulate Monitor, Thermo Fisher Scientific Inc., USA) was used to measure total aerosol mass concentrations. The TEOM was operated at a flow of 1.2 L min$^{-1}$ at 30 °C. The frequency of the tapered element was recorded every 6 s and used to calculate the mass concentration over the duration of the measurement. TEOM measurements agreed within 1%-4% with the reference (manual) gravimetric method.

Thermal-optical analysis was performed in order to establish the composition of the soot. Aerosols were sampled on three sets of two superimposed filters for each measurement point according to path b) in Fig. 1. For the coated samples, the aerosol was passed through an activated charcoal denuder first. During sampling the filters (47 mm QR-100 Quartz fibre filters, Advantec, Japan, prebaked at 500 °C for 1.5 hours) were placed in a metallic filter holder (Merck Millipore, Germany). Punches of 1.5 cm$^2$ were later used for thermal-optical analysis with a Lab OC-EC Aerosol Analyser (Sunset Laboratory Inc., Hillsborough, USA). This instrument distinguishes carbonaceous material into EC and OC (elemental and organic carbon) after being calibrated against solutions of glucose at different concentrations. The EUSAAR2-protocol (Cavalli et al., 2010) was slightly modified by extending the last temperature step to ensure that the evolution of carbon is complete (Ess and Vasilatou, 2019). OC and EC masses were determined from the upper filter, with OC masses then corrected by subtracting the mass of OC from the lower filter, consisting of the absorbed gas phase (Mader et al., 2003; Moallemi et al., 2019). The results of the thermal-optical analysis were then used to calculate *EC/TC* and *OC/TC* ratios, where *TC = OC + EC*.

## 3 Results and discussion

Two series of test aerosols were generated as summarized in Table 1. Each series consisted of four test aerosols: uncoated soot and soot with three different amounts of SOM coating. Note that the uncoated soot particles are "fresh" soot particles generated by the miniCAST burner (and not soot particles that have been coated and denuded). With Setup 1, i.e. no dilution unit between miniCAST and OCU, a high concentration of about $4\times10^7$ cm$^{-3}$ of soot particles was delivered to the OCU. The geometric mean mobility diameter ($GMD_{mob}$) of the soot particles gradually decreased from 92 nm (uncoated soot) to 83 nm (coated soot; coating 3) as shown in Fig. 2a, while the EC/TC mass fraction dropped from ~90 % to ~40 % and the *SSA* increased from about 0 to ~0.2. The operation points of the miniCAST and MSC are listed in Table S1 in the supporting information. The decrease in $GMD_{mob}$, despite the considerable amount of OC condensed on the soot particles, is not surprising. As explained in Ess et al., this is due to: i) the decrease in dynamic shape factor that dominates over the increase of volume equivalent diameter and/or ii) a restructuring of the soot core during SOM condensation (see (Ess et al., 2021a) and references therein).

**Table 1. Physicochemical properties of the uncoated and coated soot particles generated in this study. The uncertainties for the $GMD_{mob}$, total concentration, *SSA* and *AAE* correspond to one standard deviation of the mean ($k$=1; 68 % confidence interval; number of measurements $n$=100– 180 for *SSA* and *AAE*, $n$=29–35 for $GMD_{mob}$ and total concentration).**

| Operation point | $GMD_{mob}$ (nm) | $SSA_{PAX,870}$ (-) | $SSA_{neph/MAAP, 632}$ (-) | $AAE^1$ (-) | $AAE^2$ (-) | EC/TC mass fraction[3] (%) | Total concentration[4] (cm$^{-3}$) |
|---|---|---|---|---|---|---|---|
| 1 – uncoated | 91.7±0.1 | 0.027±0.001 | 0.0333±0.0002 | 1.14±0.01 | 0.875±0.014 | 91±7 | 25900±300 |
| 1 – coating 1 | 86.1±0.1 | 0.052±0.001 | 0.0749±0.0003 | 1.20±0.01 | 0.984±0.009 | 65±5 | 36500±100 |
| 1 – coating 2 | 83.4±0.1 | 0.12±0.01 | 0.148±0.001 | 1.28±0.01 | 1.05±0.01 | 48±3 | 35000±100 |
| 1 – coating 3 | 83.0±0.1 | 0.18±0.01 | 0.220±0.001 | 1.29±0.01 | 1.06±0.01 | 39±3 | 35500±100 |
| 0.1 – uncoated | 88.3±0.1 | 0.0289±0.0002 | 0.0353±0.0002 | 1.17±0.01 | 0.844±0.016 | 84±8 | 26200±100 |
| 0.1 – coating 1 | 90.2±0.1 | 0.130±0.001 | 0.156±0.001 | 1.30±0.01 | 1.15±0.02 | 37±4 | 26700±200 |
| 0.1 – coating 2 | 111 ±1 | 0.497±0.001 | 0.439±0.002 | 1.46±0.01 | 1.26±0.02 | 13±1 | 29300±100 |
| 0.1 – coating 3 | 126±1 | 0.677±0.001 | 0.646±0.002 | 1.48±0.01 | 1.36±0.02 | 10±1 | 24600±400 |

[1] $AAE$ determined from the fit over all $b_{abs}$ values (370–950 nm) from the aethalometer.

[2] $AAE$ determined from the fit over the $b_{abs}$ values (532 nm, 1064 nm) from the PTAAM.

[3] The uncertainty of the corrected OC and EC masses is based on the uncertainties given by the instrument's software, calculated as the detection limit of 0.2 µg C cm$^{-2}$ plus 5 % of the carbon mass determined in the analysis for each carbon fraction. The uncertainties due to the determination of the split point were not taken into account as they could not be quantified.

[4] Measured right after the mixing volume.

On the contrary, with Setup 0.1, i.e., including a dilution of factor 10 upstream of the OCU, the $GMD_{mob}$ of the soot particles increased from 88 nm (uncoated soot) to 126 nm (coated soot) while the EC/TC mass fraction dropped from ~85 % to ~10 % and the SSA increased up to ~0.7. Due to the lower concentration of soot particles by an order of magnitude, the α-

235 pinene/eBC$_{PAX}$ mass ratio rapidly increased to ~500 (see Table S1). As a result, the increase in volume equivalent diameter due to the high amount of condensed SOM dominated over the decrease of shape factor. The mobility size distributions of the test aerosols are displayed in Fig. 2b.

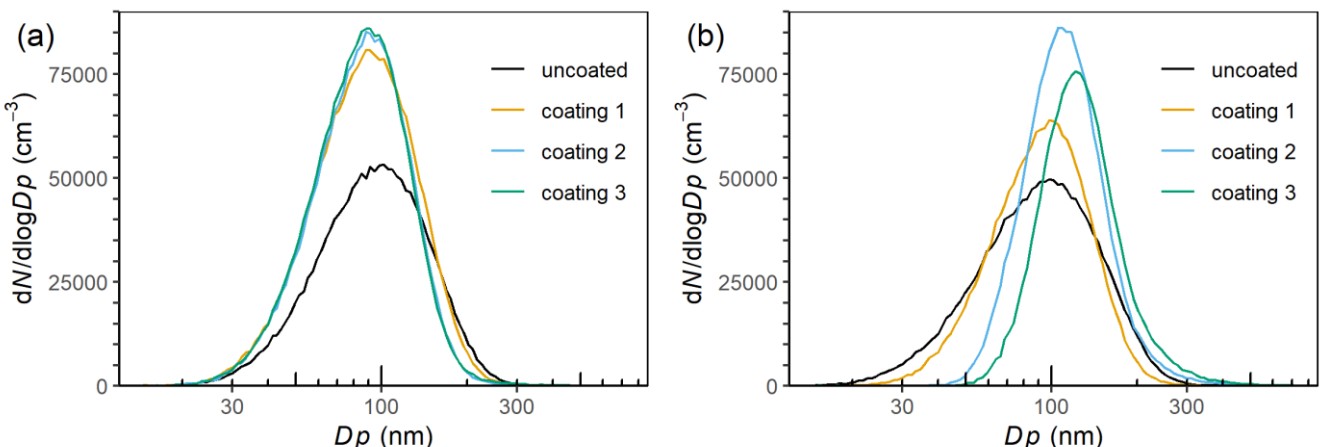

**Figure 2. Mobility size distributions as measured by SMPS (a) with Setup 1 and (b) with Setup 0.1.**

The response of the BC- and absorption-measuring instruments to the test aerosols generated by Setup 0.1 is displayed in Fig. 3. In Fig. 3b, the absorption coefficient at 532 nm ($b_{abs, 532}$) is plotted as a function of $R_{BC}$. $R_{BC} = (M_{total}-M_{BC}) / M_{BC}$ is equal to the mass of organic coating over the mass of uncoated soot as measured by the TEOM. Note that our definition of $R_{BC}$ is similar but not identical to the definition provided by Cappa et al. and Liu et al. who calculate $R_{BC}$ as [NR-PM$_{BC}$]/[BC], with NR-PM$_{BC}$ being the fraction of non-refractory particulate matter (NR-PM) exclusively associated with BC based on measurements with soot particle-aerosol mass spectrometry (Cappa et al., 2012; Liu et al., 2015). To facilitate comparison with previous literature, the total mass to BC mass ratio ($M_{total}/M_{BC}$) as measured by the TEOM is shown on the secondary x-axis. The measurements by the TEOM do not agree so well with the results from the thermal-optical analysis. We believe that this is due to the high measurement uncertainties of the thermal-optical analysis and particularly with the difficulty to define the split point. All $b_{abs}$ values have been converted to a wavelength of 532 nm using the absorption Ångström exponents determined from the fit over the two $b_{abs}$ values from the PTAAM. The absorption enhancement at 532 nm ($E_{abs, 532}$) is shown in Fig. 3c, and is equal to $b_{abs}$ of the coated soot divided by that of the uncoated soot. The EC/TC mass fraction and $SSA$ of the test aerosols are displayed in Fig. 3a (main and secondary y-axis, respectively) while $GMD_{mob}$ is shown as label on the data points.

As shown in Fig. 3b, significant deviations in the response of the different BC- and absorption-measuring instruments are observed even for the uncoated soot aerosol. Instruments based on photoacoustic spectroscopy and interferometry report a $b_{abs}$ in the range 20 to 50 Mm$^{-1}$ while the MAAP and AE33 report ~60 Mm$^{-1}$ and ~110 Mm$^{-1}$, respectively. The largest deviation is observed between the PAX and the AE33, with the AE33 overestimating $b_{abs}$ by a factor of about 2 compared to the PAX. In general, $b_{abs}$ increases with increasing SOM coating, apart from the PAS which shows a rather erratic behaviour (see Table S2 for the PAS data). The deviation between the AE33 and PAX increase with increasing SOM coating up to a factor of 3 for the "thickest" coating ($SSA \approx 0.7$, see Table 1 and Fig. 3a). Even when taking into account the expanded measurement uncertainties ($k$=2; 95% confidence interval), the measurements by the AE33 hardly agree with the measurements by the PAX and PTAAM. This indicates that the ~20% measurement uncertainty ($k$=1) assigned to the AE33 (see section 2.2) might be underestimated. Similar observations can be made for the MAAP at high $R_{BC}$ ratios even though the deviations from the PAX and PTAAM are less pronounced.

In the visible and near UV region of the spectrum, the values of $E_{abs}$ can include effects of both "lensing" and potential absorption by SOM. Absorption by α-pinene-derived SOM is very low with a MAC below 0.25 m$^2$g$^{-1}$ (Nakayama et al., 2010) or even 0.01 m$^2$g$^{-1}$ (Lambe et al., 2013) at 532 nm, depending on the oxidation state and experimental details. Instruments measuring in the wavelength region 520–637 nm all recorded an increase in $E_{abs, 532}$ as a function of $R_{BC}$ (Fig. 3c). At $R_{BC} \approx$

3.4, corresponding to an EC/TC mass fraction of 10 % and an SSA of about 0.7, an absorption enhancement in the range 1.3 (PTAAM 532 nm) to ~ 2 (MSPTI 532 nm) was observed.

A weak absorption enhancement of about 1.1-1.3 at 532 nm was calculated from the PAX data (Figure 3c). We therefore interpret the absorption enhancement shown in Figure 3c to be due to a transparent coating by SOM on the absorbing BC core, as described by (Lack and Cappa, 2010). Moreover, as biogenic SOM is only expected to absorb light in the UV and near UV region (Nakayama et al., 2010; Song et al., 2013), it is surprising that the MAAP indicates such a pronounced absorption enhancement at 637 nm. Apart from the lensing effect, one additional reason could be coating of BC in the filter by SOM or modification of the filter matrix optical properties by SOM (Lack et al., 2008). The uncertainties in Figure 3c were calculated as the quadratic sum of the uncertainties in $b_{abs}$ for the uncoated and coated soot. Note that this procedure is only a simplistic approximation. Ideally, the uncertainty in $b_{abs}$ should be partitioned in type A (random) and type B (systematic) uncertainties and correlations between the different components should be taken into account. A robust uncertainty calculation was, however, not possible because the uncertainties of the instruments are not so clearly understood and, additionally, instruments such as the PAS and the MSPTI at times suffered from unexpected technical errors. In the case that $b_{abs}$ is dominated by systematic uncertainties which remain the same when measuring the uncoated and coated soot particles, such uncertainties may cancel out, resulting in a much smaller combined uncertainty in $E_{babs}$ than what presented in Figure 3c.

A limitation of our study is that the relative humidity of the uncoated soot aerosols when entering the organic coating unit was low to very low (see subsection 2.1). No experiments were performed at high RH due to the presence of homogeneously nucleated SOA particles at RH above 40%-50%. It is known that the absorption by BC depends strongly on RH, and may lead to a factor of two increase in absorption at high RH compared with dry conditions (Fierce et al., 2016). This is one of the reasons why GAW (Global Atmospheric Watch) recommends measurements of light absorption at low RH (GAW, 2016). Moreover, soot-containing particles generated in the laboratory under controlled conditions might have a more uniform composition compared to the aged soot particles in ambient air. In general, the range of absorption enhancement $E_{abs} = 1.1$-1.3 calculated based on the results by the PAX and PTAAM agrees very well with (Fierce et al., 2016) who calculated a limited absorption enhancement ($E_{abs} = 1$-1.5) at low RH, when accounting for particle-level variation in composition of soot-containing particles. The dry conditions during and after ozonolysis of α-pinene may have also had an effect on the phase state of the SOM, leading most probably to a solid-state coating (Saukko et al., 2012).

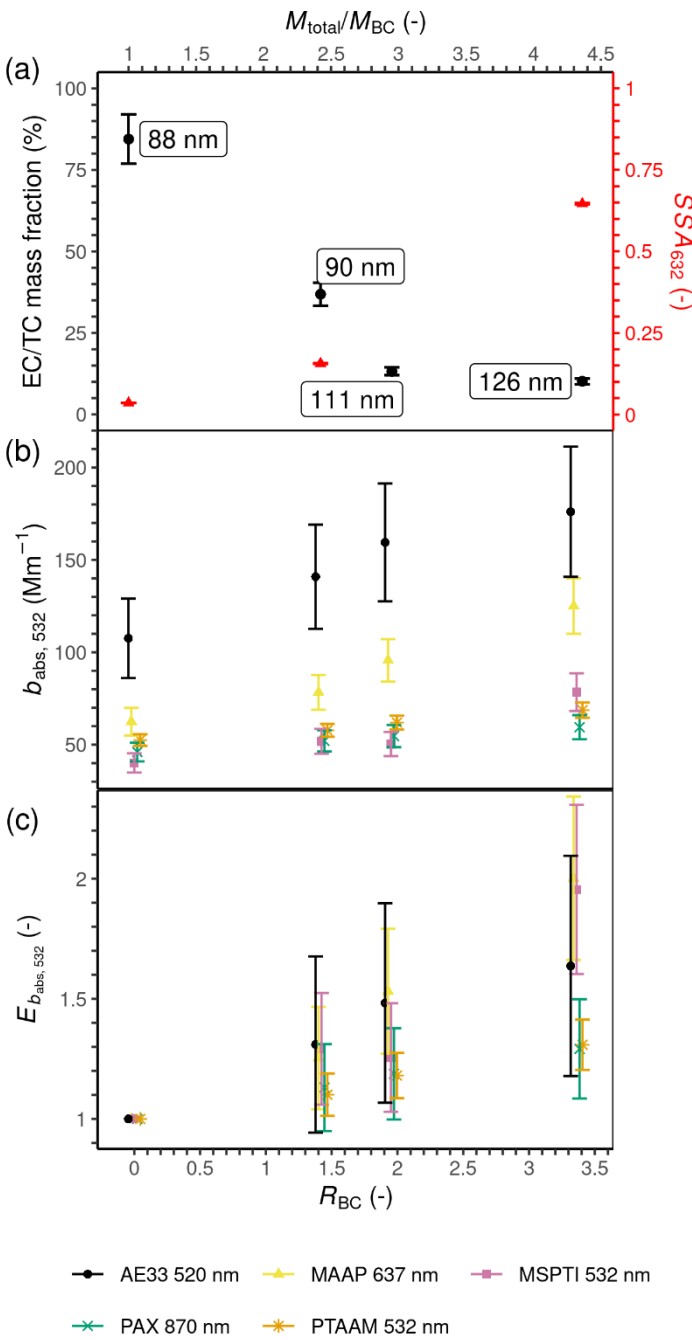

**Figure 3. Measurements obtained with Setup 0.1. (a) EC/TC mass fraction as a function of total mass to BC mass ratio and $R_{BC}$. The**
**total mass to BC mass ratio and $R_{BC}$ are based on TEOM measurements. The single-scattering albedo (*SSA*) at 637 nm, calculated**

from $b_{abs}$ measured by the MAAP and the scattering coefficient measured by the nephelometer, is shown on the secondary y-axis. The sample geometric mean diameter is shown as a label on each data point. (b) Absorption coefficient ($b_{abs}$) as a function of total mass to BC mass ratio and $R_{BC}$. All $b_{abs}$ values have been converted to a wavelength of 532 nm using the absorption Ångström exponents determined from the $b_{abs}$ values of the PTAAM. The legend below Figure 3c indicates the wavelengths at which the measurements were performed. All $b_{abs}$ have also been scaled by the number concentration. The values are listed in Table S2. (c) Absorption enhancement factor $E_{b_{abs}}$(532 nm) as a function of total mass to BC mass ratio and $R_{BC}$. The data points in panels b) and c) have been slightly shifted along the x-axis to improve readability of the graph. The error bars designate measurement uncertainties for $k$=1 (68 % confidence interval). The uncertainty ($k$=1) in $R_{BC}$ is estimated to be about 5% (not shown).

Figure 4 shows the response of the BC- and absorption-measuring instruments to the test aerosols generated by Setup 1. In this case, coating is more moderate and, as explained above, the $GMD_{mob}$ of the soot particles decreases slightly upon coating. AE33 overestimates $b_{abs}$ by up to a factor of 2 to 3 compared to the other instruments as shown in Fig. 4b. The instruments report no or only a weak absorption enhancement as a function of SOM coating (Figure 4c). During the measurement campaign the PAS was affected by a number of technical issues, which intermittently caused high uncertainties in the measurements. As a result, the PAS data are not shown in Figure 4 but can be found in Table S3.

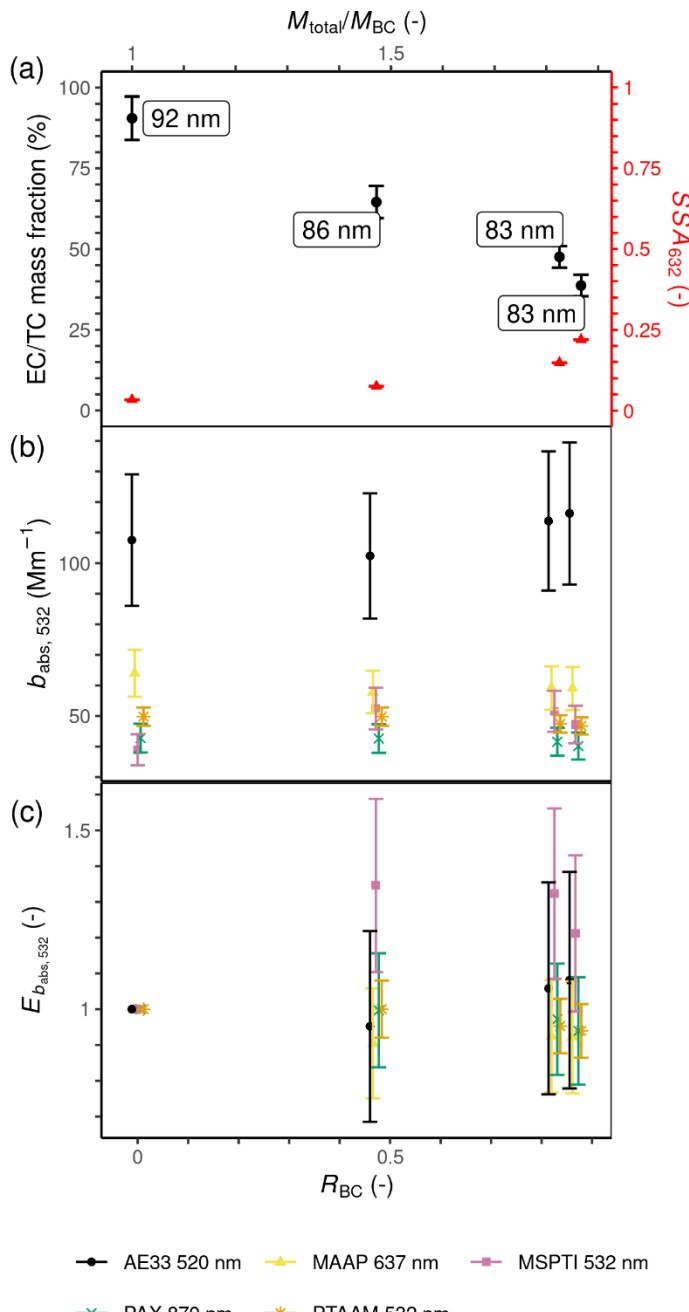

**Figure 4: Measurements obtained with Setup 1. (a)** EC/TC mass fraction as a function of total mass to BC mass ratio and $R_{BC}$. **The total mass to BC mass ratio and $R_{BC}$ are based on TEOM measurements. The single-scattering albedo ($SSA$) at 637 nm, calculated from $b_{abs}$ measured by the MAAP and the scattering coefficient measured by the nephelometer, is shown on the secondary y-axis. The sample geometric mean diameter is shown as a label on each data point. (b)** Absorption coefficient ($b_{abs}$) as a function of total

**mass to BC mass ratio and $R_{BC}$. All $b_{abs}$ values have been converted to a wavelength of 532 nm using the absorption Ångström exponents determined from the $b_{abs}$ values from the PTAAM. The legend below Figure 4c indicates the wavelengths at which the measurements were performed. All $b_{abs}$ have also been scaled by the number concentration. The values are listed in Table S2. (c) Absorption enhancement factor $E_{b_{abs}}$(532 nm) as a function of total mass to BC mass ratio and $R_{BC}$. The values are listed in Table S3. The data points in panels b) and c) have been slightly shifted along the x-axis to improve readability of the graph. The error bars designate measurement uncertainties for $k=1$ (68 % confidence interval). The uncertainty ($k=1$) in $R_{BC}$ is estimated to be about 5% (not shown).**

Two photo-thermal instruments based on different designs (MSPTI and PTAAM) were operated in parallel to measure the aerosol absorption coefficient. Here, we compare the measurements performed at 532 nm. The response of the PTAAM was regularly tested during the campaign and showed average variation of 3% for the 532 nm channel (Supplement S6). Testing of the MSPTI response showed larger variability at the end of the measurement campaign as the laser became more unstable. This especially affected the measurements of the uncoated particles, which were performed at the end; due to this fact the MSPTI to PTAAM ratio for the uncoated particles is more uncertain (Fig. S6a). Two additional one-day experiments were performed comparing different coating treatments (Figure S6b). These measurements show the opposite behaviour for the uncoated particles compared to Figure S4a. Comparing the experiments one can conclude that the average response of both instruments agree well within the measurement uncertainty, thus showing similar absorption enhancement at 532 nm for both instruments.

To decouple a possible "lensing" effect from the light absorption by SOM, the absorption enhancement in the near infrared (NIR) region, $E_{abs, 950}$ is plotted as a function of $R_{BC}$ in Fig. 5 (for Setup 0.1). Biogenic SOM does not absorb in the NIR region (Nakayama et al., 2010; Schnaiter et al., 2003; Xie et al., 2017); thus, any absorption enhancement would be due to the "lensing" effect. In this study, the only instruments measuring in the NIR were the AE33 (950 nm), the PAX (870 nm) and the PTAAM (1064 nm). Prior to the calculation of $E_{abs}$, all $b_{abs}$ values had been converted to a wavelength of 950 nm using the absorption Ångström exponents determined from the pair of $b_{abs}$ values at 880 nm and 950 nm reported by the aethalometer. As shown in Fig. 5, the measurements by the PTAAM and PAX agree very well and both instruments yield an $E_{abs, 950}$ close to 1. On the contrary, the AE33 reports an absorption enhancement as a function of the organic coating, with $E_{abs, 950} \approx 1.5$ at $R_{BC} \approx 3.4$. It is known that the multiple-scattering parameter $C$ of the aethalometer depends on the $SSA$ and, possibly, on the size of the aerosol particles (Yus-Díez et al., 2021). As mentioned earlier, this variation was not taken into account but, instead, a fixed $C$ value of 1.39 (provided by the manufacturer) was applied throughout this study to obtain the absorption coefficient from the measured attenuation coefficient. We believe that the absorption enhancement reported by the aethalometer is an artefact arising from keeping the $C$ value fixed. Under this assumption, it is possible to calculate new values of $C$ (Bernardoni et al., 2021) as a function of $SSA$ using the mean $b_{abs}$ value of the PTAAM and PAX as a reference:

$$C_{SSA} = \frac{b_{atn,AE}}{b_{abs,ref}}$$

(2)

This results in calculated values of $C$ at 950 nm of 3.4, 4.6, 4.9 and 5.3 for $SSA_{870}$ values (measured by PAX) of 0.03, 0.13, 0.50 and 0.68 respectively.

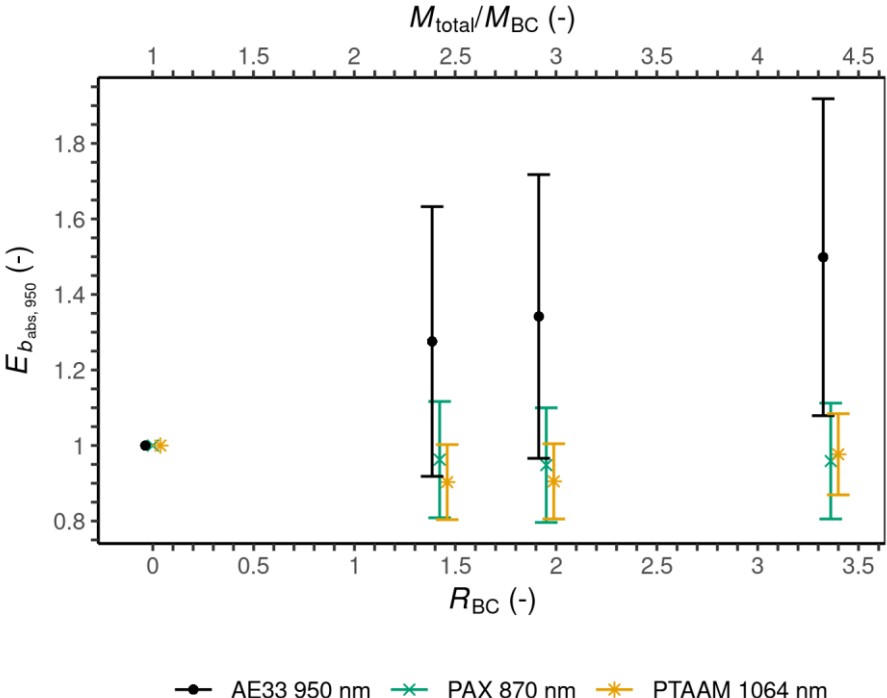

**Figure 5: Measurements obtained with Setup 0.1. Absorption enhancement factor $E_{abs}$ (950 nm) as a function of total mass to BC mass ratio and $R_{BC}$. For the calculation of $E_{abs}$, all $b_{abs}$ values had been converted to a wavelength of 950 nm**
**using the absorption Ångström exponents determined from the 880 nm and 950 nm $b_{abs}$ values from the aethalometer. All $b_{abs}$ had also been scaled by the number concentration. The values are listed in Table S4. The data points have been slightly shifted along the x-axis to improve readability of the graph. The error bars correspond to uncertainties for $k=1$ (68 % confidence interval; see text for more details).**

If the $SSA$ of the ambient aerosol is known, then the calculated $C_{SSA}$ values can be used to post-correct the data from the
365 aethalometer. In well-equipped monitoring stations, this can be performed using an integrating nephelometer and a reference absorption measurement at a single wavelength (Yus-Díez et al., 2021), and can be extended using a multi-wavelength reference absorption measurement and a multi-wavelength scattering measurement in a representative campaign. Multi-wavelength absorption corrections and the determination of $C$ can be derived from off-line filter measurements with a time resolution of hours to days (Bernardoni et al., 2021). Moreover, a recent study suggests that "the low-cost and widely used PA
monitors can be used to measure and predict the aerosol light scattering coefficient in the mid-visible nearly as well as

integrating nephelometers" (Ouimette et al., 2021). By letting a low-cost nephelometer (temperature- and RH-controlled) run parallel to the AE33 at monitoring stations, an approximate SSA value (based $b_{abs}$ by the AE33 and $b_{scat}$ by the low-cost sensor) can be calculated and the $b_{abs}$ of the AE33 can be refined by implementing a new $C_{SSA}$. Both $b_{abs}$ by the aethalometer and $SSA$ can then be refined in multiple steps in an iterative procedure.

## 4 Conclusions

A series of test aerosols were produced using a miniCAST BC generator and a novel organic coating unit, comprising of a micro smog chamber and an integrated dosing system for VOC. Both uncoated soot particles and soot particles coated with varying amounts of α-pinene derived SOM were generated covering a wide range of particle sizes (83–126 nm), EC/TC mass fractions (10–91 %) and optical properties ($SSA$ almost 0 to 0.7 at 637 nm).

Several BC- and aerosol-absorption-measuring instruments were compared using these aerosols: A dual-spot aethalometer, a photoacoustic extinctiometer (PAX, 870 nm), a MAAP, a prototype photoacoustic instrument and two prototype photothermal interferometers (PTAAM-2λ and MSPTI). This is the first time that the PTAAM-2λ, which is now commercially available, has been compared to other absorption-measuring instruments. In general, the filter-based instruments (AE33 and MAAP) overestimated $b_{abs}$ compared to in situ measuring instruments. The bias is systematic and increases with increasing SSA. The absorption enhancement is equally highest for the filter-based instruments. The PAX and the NIR channel of the PTAAM measured almost no enhancement, though a weak absorption enhancement was observed at 532 nm.

The setup of miniCAST combined with the novel organic coating unit and the methodology described in this study provide a straightforward and reproducible procedure for the inter-comparison and characterisation of both filter-based and in situ BC-measuring instruments. The system is very robust, compact, relatively inexpensive and allows to generate realistic test aerosols in a reproducible and standardized manner. Additionally, in comparison with smog chambers, stability of the aerosols is reached within minutes after changing operation points, allowing for several measurements within a day.

A limitation of this study was that the relative humidity of the uncoated soot aerosols when entering the organic coating unit was below 30%. More studies are needed at higher relative humidity in order to better simulate atmospheric processes.

**Data availability**

Measurement data will be made available for the final publication at https://zenodo.org/communities/aerotox/ .

**Author contribution**

DMK and KV designed the study and wrote the manuscript with contributions from all authors. DMK generated the model aerosols and analysed the data of the AE33 and PAX, GM and LD operated and analysed the data of the PTAAM, BV and JR operated and analysed the data of the MSPTI, MO operated and analysed the data of the PAS, APH analysed the data of the

MAAP and nephelometer. All authors contributed to the interpretation of the results.

**Competing interests**

LD and GM are (in part) employed by the manufacturer of the PTAAM-2λ.

**Acknowledgments**

DMK and KV thank Dr. Michaela Ess (previously at METAS) for valuable technical support during the preparation of the measurement campaign.

**Funding**

This work has received funding from the EMPIR 18HLT02 AeroTox and 16ENV02 Black Carbon projects. EMPIR is co-financed by the Participating States and from the European Union's Horizon 2020 research and innovation programme. This

research has also been supported by the Swiss National Science Foundation (grant no. 200021_172649) and the EUROSTARS programme (IMALA, grant no. 11386).

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
