# Peer review of "Comparing black carbon and aerosol absorption measuring instruments – a new system using lab-generated soot coated with controlled amounts of secondary organic matter"

_Atmospheric Measurement Techniques, 2021_

## Author Comment (AC1)

We would like to thank Reviewer 1 for the feedback, which has helped us to improve the manuscript. Below we added the response to each comment/question as well as the sections of manuscript that were adapted.

**Reviewer**: Page 1. Title. It seems that the title captures only a part of the accomplishment of this work. As stated in the conclusions, one outcome is the demonstration of the aerosol generation system. Perhaps the title could be revised to reflect this.

> **Response**: Thank you for this comment. We have revised the title to "Comparing black carbon and aerosol absorption measuring instruments – a new system using lab-generated soot coated with controlled amounts of secondary organic matter".

**Reviewer**: Page 5, paragraph starting at line 145. One goal of this study seems to be to evaluate new instruments with existing instruments. Since the 'PAS' is a new instrument, it would be good to give more details and a schematic. It seems that operation at such a high frequency (around 23 kHz) would result in a low pass filter response to light absorption, that heat transfer from the aerosol to the surrounding air would not all occur within the acoustic period so that the measurements would be strongly a function of aerosol size. Additional details about the motivation for, and design of, the prototype PAS would be useful, perhaps given in the supplemental section. The additional detail could explain how the instrument 'modulation frequency adapts…' (what measurements or theory are used to accomplish this).

> **Response**: Thank you for pointing this out. We have added a detailed description of the PAS to the Supporting Information in Section S4 as follows:

**S4 Description of the PAS**

The measurement principle of the PAS is based on the photo-thermal effect of light absorbing particles in an acoustic resonator, as shown in Fig. S3. BC particles in the laser beam absorb light (1), which causes them to heat up (2). This heat is transferred to the carrier gas, releasing a pressure wave from the particle (3). This effect is enhanced if the light is modulated at the right frequency (4), building a standing wave within the resonator which is measured with a sensitive microphone (5). By matching the light intensity modulation frequency with the resonance frequencies of the chamber, the quality factor of the acoustic mode ($Q \sim 1000$ at 22.7 kHz) enhances the signal amplitude. The amplitude corresponds linearly to the amount of absorbed light.

[Figure]

**Figure S3. Illustration of the photothermal effect.**

The photoacoustic instrument uses a novel resonator chamber with elliptical cross-section to enhance the photoacoustic, three lasers, a microphone, a loudspeaker, an amplifier and a signal-processing unit (lock-in amplifier). The three different wavelengths (445 nm with 700 mW, 520 nm with 300 mW, 638 nm with 300 mW power) of the diode-lasers allow us to measure wavelength-dependent optical properties of the aerosols.

The motivation for the elliptical cross-section is the possibility to separate the attenuation position from the measurement position of the microphone - at approximately the two focal points of the ellipse. The laser beam is guided by mirrors into the aluminium resonator along a focal point, which allows effective excitation of the transversal modes. The microphone is situated in the mid of the 24 cm long resonator case and can be moved transversally into the resonator, as indicated in Fig. S4. The microphone signal is then amplified and demodulated with a Stanford SR850 lock-in amplifier, which allows to measure amplitude and response delay of the photoacoustic signal at the excitation frequency, working currently at around 22.7 kHz. The measurements were performed with an integration time of 1sec therewith the mode can easily build up to a standing wave. The laser intensity modulation frequency is adapted periodically every 10 minutes, with the help of a loudspeaker and a frequency sweep, to match with the resonance frequency of the chamber.

[Figure]

**Figure S4: Schematic illustration of the PAS.**

A loudspeaker (not shown in Fig. S4) is guided into the ellipse (parallel to the microphone) and attenuates the acoustic modes of the resonator chamber. These modes are independent of the light-absorbing particles in the chamber, but depend on the gas composition, temperature and pressure. The frequency of the loudspeaker is swept within a window where an acoustic resonance occurs. The frequency of the amplitude peak of this spectrum is then determined and its relative shift (to a reference measurement) is added to the laser intensity modulation frequency.

The motivation for the high frequency is on one hand to explore the response of the photoacoustic in the ultrasonic regime, as this was little explored in the past, and on the other hand to eventually measure the aerosol size distribution of the sample. At higher modulation frequencies it is easier to measure the phase delay between different aerosol size distributions, as the delay gets bigger for higher frequencies. At a fixed frequency, larger particle distributions could show a bigger phase delay then small particles, as the heat capacity rises with particle size. This phase delay was not yet measured and is still subject of investigation. Also measurements in the ultrasonic regime are expected to be less influenced by ambient acoustic noise.

**Reviewer**: Page 6, line 180. It's unclear to me how the babs values were normalized by total number concentration. Total number concentration seems potentially fraught with issues since light absorption of course depends on particle diameter in a complicated way.

**Response**: The total concentration is shown in the last column of Table 1.

| Operation point | $GMD_{mob}$ (nm) | $SSA_{PAX,870}$ (-) | $SSA_{neph/MAAP}$ (-) | $AAE^1$ (-) | $AAE^2$ (-) | EC/TC mass fraction[3] (%) | Total concentration[4] ($cm^{-3}$) |
|---|---|---|---|---|---|---|---|
| 1 – uncoated | 91.7±0.1 | 0.027±0.001 | 0.0333±0.0002 | 1.14±0.01 | 0.875±0.014 | 91±7 | 25900±300 |
| 1 – coating 1 | 86.1±0.1 | 0.052±0.001 | 0.0749±0.0003 | 1.20±0.01 | 0.984±0.009 | 65±5 | 36500±100 |
| 1 – coating 2 | 83.4±0.1 | 0.12±0.01 | 0.148±0.001 | 1.28±0.01 | 1.05±0.01 | 48±3 | 35000±100 |
| 1 – coating 3 | 83.0±0.1 | 0.18±0.01 | 0.220±0.001 | 1.29±0.01 | 1.06±0.01 | 39±3 | 35500±100 |

GMDmob (see column 2) does not change much throughout the experiment, therefore any artefacts arising from the dependence of light absorption on particle size are expected to be negligible.

Example:

For "Operation point 1", the concentration of uncoated particles was taken as a reference value. The b_abs values of the coated points were adjusted by a correction factor equal to the ratio of the concentrations. For instance, for the first coating, b_abs was multiplied by 25900/36500 = 0.71.

**Reviewer**: Page 8, line 215. Was electron microscopy performed to investigate soot core restructuring?

**Response**: This is indeed an important point. Cryo-TEM images of soot and coated soot particles were recorded during the study by (Ess et al., 2021) doi.org/10.1016/j.jaerosci.2021.105820.

The left panel below shows the TEM image of an uncoated (bare) soot particle, the middle panel is the image of a partly coated soot particle and the right panel the structure of a fully coated (i.e. embedded in the organic matter) soot particle. A gradual collapse of the soot core is observed as more SOA condenses on the soot core.

[Figure]

[Figure]

[Figure]

Our findings agree well with those reported by Wang et al. in the study "Fractal Dimensions and Mixing Structures of Soot Particles during Atmospheric Processing". Environmental Science & Technology Letters 2017 4 (11), 487-493, DOI: 10.1021/acs.estlett.7b00418 (see figure below):

[Figure]

**Reviewer**: Page 9, line 247. It seems that there is a contradiction in this paper to use the AE33 derived AAE to move the babs for other sensors (PAX for example) values all to 532 nm since the AE33 data is also described as being likely incorrect. What is the justification for doing this and what uncertainty enters the conclusions as a consequence?

>**Response**: This is a valid concern. Figures 3 and 4 show b_abs at 532 nm. Two of the sensors (MSPTI and PTAAM) measured already at 532 nm, so no correction was applied. Three other sensors (AE33 520 nm, PAS 520 nm and MAAP 670 nm) measured at a wavelength close to 532 nm, therefore only a very small correction was needed. Any artefacts related to this correction are expected to be negligible compared to the systematic uncertainties of these instruments. The PAX is the only instrument that measured at 870 nm and would require a considerable correction for converting b_abs to 532 nm.

>In Table 1, two sets of AAE values are calculated based on measurements by the AE33 and by the PTAAM. AAE as calculated based on AE33 is about 20 % higher than the AAE based on PTAAM. This difference is most probably a result of the artefacts related to the interaction between the particles and the filter matrix. Since the sample aerosol features very uniform diameters (Table 1), we believe the reason for this difference is the wavelength dependence of the multiple scattering parameter $C$ in the AE33. The use of the AE33 derived AAE overestimates systematically the extrapolated absorption coefficients (from 870 nm to 532 nm) for PAX. Compared to the calculation using the PTAAM AAE, this overestimation is 10%-14%. We have now revised Figs 3 and 4 of the manuscript by calculating b_abs based on AAE values from the PTAAM. We have also revised Tables S2 and S3 accordingly.

**Reviewer**: Page 9, lines 254-257. Which instrument(s) is (are) being considered reference methods for evaluating the results of these measurements? Which instruments provide the most and least correct measurements and how is that known?

>**Response**: We believe that the AE33 is the instrument that suffers from the largest systematic errors. As discussed in this paper and in many previous publications, aethalometers assume a constant multiple scattering parameter $C$, which is not correct. $C$ depends on the properties of the filter, design of apparatus and on the properties of the sample aerosol (e.g. particle size and SSA).

>In our opinion, the PTAAM is probably the best candidate instrument to serve as reference method since the instrument can be calibrated at 532 nm with $NO_2$. The calibration is transferred to the infrared (IR) channel using aerosolized nigrosin and its relative green-to-infrared absorption ratio determined using a Mie calculation based on size distribution measurements. However, since the PTAAM is a very recent development and the manuscript on the design and calibration of the instrument is still in preparation, we felt it might be premature to make such a strong statement in the current manuscript. Moreover, dedicated inter-comparisons between PTAAM and PAX are needed to understand the deviations between the two instruments. Such studies are already in progress and the results will be presented in due time.

**Reviewer**: Page 9, line 257, and later in the paper, page 10, lines 264-267. It still is not clear why the PAX 870 nm absorption measurements show no sensitivity to coatings. At what wavelengths are the coatings expected to be light absorbing to various degrees? It seems that the authors could readily evaluate the insensitivity to coating at 870 nm by using the

core/shell model for light absorption as a function of wavelength to theoretically evaluate light absorption enhancement at various wavelengths to confirm these results. Mass transfer is known to affect photoacoustic/photothermal measurements when the aerosol (or coating) has a high vapor pressure.

**Response**: This is an interesting and complex topic. There are two issues: first, the wavelength dependence of absorption of coatings (or coated absorbing cores); second, the "loss" of the latent heat from the photo-thermal or photoacoustic signal due to the volatilization of coatings.

**Wavelength dependence**

In this study, the soot coatings consisted of secondary organic matter from the oxidation of a biogenic precursor (α-pinene). According to the literature, biogenic SOA does not absorb as strongly as anthropogenic SOA. Light absorption is only observed in the UV and near-UV region:

https://www.ncbi.nlm.nih.gov/pmc/articles/PMC7372712/

https://doi.org/10.1016/S0021-8502(03)00361-6

Under certain experimental conditions, SOA from the ozonolysis of α-pinene showed negligible light absorption even at near-UV wavelengths:

https://doi.org/10.1029/2010JD014387

Therefore, it is not surprising that the PAX at 870 nm did not show any sensitivity to coatings. We also see the increase of the AAE with increasing thickness of the coating in the PTAAM (Table 1). We interpret this as the increase in absorption at lower wavelengths (532 nm) relative to the higher wavelengths (IR). Additionally, we see with PTAAM an enhancement of absorption at 532 nm of about a factor of 1.3 (Fig. 3) and no enhancement at 1064 nm (Fig. 5).

There are some differences between different modelled results regarding this effect. Virkkula (https://doi.org/10.5194/amt-2020-438) states that lensing effect is less pronounced at longer wavelengths using a Mie model. Liu et al. (http://dx.doi.org/10.1016/j.jqsrt.2015.08.005) show (Figure 9) that the calculated lensing effect differs between the discrete-dipole approximation (DDA) and core-shell Mie models. The DDA model predicts lower lensing at longer wavelengths; core-shell model predicts a similar magnitude of lensing from the visible to the IR.

The differences between different models can be reconciled with measurements which we present using PTAAM.

**Volatilization of coatings**

We have evaluated the volatility of our SOA coating using a total carbon analyzer developed by FHNW (manuscript in preparation). We do not have any evidence that the SOA coatings used in our study are more volatile than typical suburban aerosols. Had the coatings we produced been volatile, this would have been seen in the comparison between the PAX and PTAAM.

PTAAM measures absorption at 532 nm and 1064 nm of the same sample (retention time for aerosol in the pump beam is 1000-times longer compared to the modulation

interval). In principle, it would be possible for the pump beam of the PTAAM to evaporate the soot coating. Increase of the measured AAE shows, however, that at least part of the coating has not been removed. Estimated pump beam intensity in PTAAM is 2 W/mm2 (PTAAM article in preparation – will be submitted shortly).

The recondensation of the possibly volatile coatings would occur during the off-phase of the pumping laser in the PAX and PTAAM. In the PAX, the acoustic wave generated by absorption of light, moves away from the sample signal generation region by the time recondensation occurs. In the photo-thermal interferometer, the signals due to volatilization and recondensation are generated in the region probed by the probe beam, so the recondensation is measured and the latent heat is taken into account properly. We do not expect any absorption by the coating itself and no absorption enhancement by the coated absorbing cores (see arguments above). We would expect to measure the difference in the IR between the PAX and PTAAM if the SOA coatings had been volatile. We saw no differences (Fig. 5) in agreement with our total carbon measurements.

**Reviewer**: Collectively the results shown in Figures 3b and 4b indicate that common aerosol light absorption measurement methods disagree by a factor of two, even in this case of laboratory-generated aerosol. If the measurements were given error bars that indicate precision and accuracy, would they overlap?   It seems that the AE33 results are farthest from the rest, and that may be due to the choice of factors used to convert from filter-based to in situ light absorption.   Assuming that the photoacoustic/photothermal measurements are closest to reality, can these measurements provide a suggested correction strategy for the AE33 light absorption calculation as a contribution to a growing literature on the subject?

**Response**: This is a very good question. Filter-based methods clearly suffer from systematic errors, which are much larger than the statistical ones. At the moment, we cannot give error bars for the combined uncertainties because the systematic errors can only be quantified by comparison to a reference method. Two methods which could serve as reference are the "extinction minus scattering method (EMN)" and photo-thermal interferometry (note that EMN suffers from large uncertainties at high aerosol SSA).

A correction strategy for the AE33 light-absorption calculation is discussed on pages 14-15 (Figure 4). We suggest, as others have done before us, to calculate new values for the aethalometer's $C$ value as a function of SSA using the babs value of the PTAAM (or PAX) as a reference:

$$C_{SSA} = \frac{b_{atn,AE}}{b_{abs,ref}}$$

If the SSA of the ambient aerosol is known, then the calculated C(SSA) value can be used to post-correct the AE33 data. In well-equipped stations, this can be performed using an integrating nephelometer and a reference absorption measurement at a single wavelength (Yus-Diez et al., 2021, https://doi.org/10.5194/amt-2021-46), and can be extended using a multi-wavelength reference absorption measurement and a multi-wavelength scattering measurement in a representative campaign. Multi-wavelength absorption corrections and the determination of $C$ can be derived from off-line filter measurements with a time resolution of hours to days (Bernardoni et al., 2021, https://doi.org/10.5194/amt-14-2919-2021). Moreover, a recent study suggests that "the low-cost and widely used PA monitors can be used to measure and predict the aerosol light scattering coefficient in the mid-visible nearly as well as integrating

nephelometers" (Ouimette et al., 2021, https://doi.org/10.5194/amt-2021-170). By letting a low-cost nephelometer run parallel to the AE33 at monitoring stations, an approximate SSA value (based on AE33 b_abs and low-cost sensor b_scat) can be calculated and the babs of the AE33 can be refined by implementing a new C(SSA). Both b_abs by the aethalometer and SSA can be then refined in multiple steps in an iterative procedure.

We have amended Section 3 to include this information.

**Reviewer**: Minor issue: Page 15 line 333, 'of' is repeated.

**Response**: Thank you for spotting this error. We have removed the second "of".

---

## Author Comment (AC2)

We would like to thank Reviewer 2 for the valuable feedback and suggestions for improvement. Please find below the response to each comment/question as well as the sections of manuscript that were modified.

Reviewer: Accurate and precise measurements of aerosol light absorption remain the primary limitation towards reducing the uncertainty in direct radiative forcing brought about To this end, the present manuscript brings together well-established black carbon. instrumentation (AE-33 Aethalometer, single wavelength MAAP and a single wavelength PAX together) with new or prototype in situ measurements (3-wavelength (445 nm, 520 nm, and 638 nm) prototype photoacoustic sensor, a dual wavelength (1064 nm and 532 nm) photothermal interferometer (PTI) and a second single wavelength (532 nm)) to intercompare and evaluate the performance of all measurement methodologies. Black carbon (BC) aerosol were generated using small burner outfitted with an oxidation chamber to produce secondary aerosols that could, in turn, coat the furnace-generated BC particles thereby enabling a comparison between uncoated and coated BC particles. By and large, the manuscript is clearly written save a few sentences - highlighted below - and the reported findings this manuscript are of value - especially the performance of the photothermal interferometers, which, have not been fully utilized towards the measurement of BCcontaining particle light absorption. This said, there are a couple of nagging issues, discussed below, that this reviewer feels needs to be addressed before acceptance for publication can be rendered.

**Reviewer:** An important caveat with the present work is that there no "direct" measurement of amount of coating mass associated with the soot particles, such as might be procured from a BC particle mixing state analysis using the Single Particle Soot Photometer (SP2) or a measurement conducted on known soot particle diameters and where a mass classifier could then be used to analyze the amount of coating on the BC particle. Instead, there is only an indirect measure of the coating mass viz-a-viz a EC/TC measurement. This may be a subtle point, but it is quite important as the authors use estimated coating to core mass ratios on polydispersed size distribution to derive conclusions about the observed absorption properties.

**Response:** Thank you for your comment. The amount of coating mass associated with the soot particles was calculated based on gravimetric measurements by TEOM (direct method). This is explained in line 240: "In Fig. 3b, the absorption coefficient at 532 nm (babs, 532) is plotted as a function of RBC. RBC = (Mtotal-MBC) / MBC is equal to the mass of organic coating over the mass of uncoated soot as measured by the TEOM". Also, in the captions of Figures 3 and 4 it is mentioned that "The total mass to BC mass ratio and RBC are based on TEOM measurements".

The Reviewer is absolutely right that additional measurements with a mass classifier would have been beneficial. The reason why we did not perform such measurements was the lack of availability of an APM or CPMA. Concerning the use of SP2 for deriving rBC mass, we believe that this method suffers from high uncertainties at such small particle sizes (GMDmob 90 nm), which are at the limit of detection. Nevertheless, such measurements have been performed in an experiment conducted by some of the co-authors and are presently being analyzed (and will be featured in a future publication).

**Reviewer:** A central issue with this manuscript is the attribution of absorption enhancement. A quick back-of-the-envelope Mie calculation, in the core-shell approximation limit, suggests that even a modest coating thickness of 25 nm of a non-absorbing coating on top a 100 nm BC core will lead to absorption enhancements of 1.4 at 550 nm, 1.3 at 870 nm, and even

1.26 at 1064 nm (and even with at 10 nm coating, the enhancements are all on the order of 1.17-1.13. So the observation of no enhancement at the longer wavelengths is a bit puzzling (unless one brings in particle diversity, which the authors do not bring up). Additionally, the authors do not discuss the potential impacts of measurement uncertainty on their analysis (in some cases the decrepancy among the in situ measurements approaches 75%). Instead, the authors suggest that the absence of observed absorption enhancement at 870 nm is consistent with that reported by Cappa et al.. This is a highly-glossed over argument. The Cappa work compared ambient urban emissions with ambient emissions that were passed though a denuded, from which absorption enhancements were derived. Even given the great care the authors exercised in the Cappa work, the authors acknowledged in a later reply, about the possibility that the denuder did not remove all the coating. So the authors are urged to exercise when using Cappa et al. observations to analyze their observations. Addiitonally, the authors are encouraged to consider the potential impacts of particle diversity on their observations (e.g., see Fierce, et al. Black carbon absorption at the global scale is affected by particle-scale diversity in composition. Nat Commun 7, 12361 (2016)). This said, given the very large discrepancies in the observations, the authors would be well served providing a discussion on the impacts of measurement uncertainty on their conclusions.

**Response:** We thank the Reviewer for this valuable comment. We were not aware that the work by Cappa et al. might have been biased. We have now removed the sentence in line 262: "This is in good agreement with Cappa et al., who reported that BC emitted from large to medium-sized urban centres (dominated by fossil fuel emissions) does not exhibit a substantial absorption enhancement when internally mixed with non-BC material (Cappa et al., 2012). Eabs during both field campaigns exhibited minimal dependence on RBC, with Eabs, 532nm remaining close to 1 (absorption was measured by photoacoustic spectroscopy)".

We amended the text (Line 279) as follows: "A limitation of this study is that the relative humidity of the uncoated soot aerosols when entering the organic coating unit was low to very low (see subsection 2.1). No experiments were performed at high RH due to the presence of homogeneously nucleated SOA particles. It is known that the absorption by BC depends strongly on RH, and may lead to a factor of two increase in absorption at high RH compared with dry conditions (Fierce et al., 2016). This is one of the reasons why GAW (Global Atmospheric Watch) recommends measurements of light absorption at low RH (GAW report no. 226, 2016). Moreover, soot-containing particles generated in the laboratory under controlled conditions might have a more uniform composition compared to the aged soot particles in ambient air. In general, the range of absorption enhancement  $E_{abs} = 1.1-1.3$  calculated based on the PAX and PTAAM agrees very well with (Fierce et al., 2016) who calculated a limited absorption enhancement ( $E_{abs} = 1-1.5$ ) at low RH, when accounting for particle-level variation in composition of soot-containing particles. The dry conditions during and after ozonolysis of  $\alpha$ -pinene may have also had an effect on the phase state of the SOM, leading most probably to a solid-state coating (Saukko, 2012)".

We also amended the section "Conclusions" (Line 390) as follows: "A limitation of this study was that the relative humidity of the uncoated soot aerosols when entering the organic coating unit was below 30%. More studies are needed at higher relative humidity in order to better simulate atmospheric processes".

When replying to a similar question from Reviewer 1, we noted that the enhancement (or lack of it at 870 nm) is dependent on two different issues. We repeat these two arguments here. It is important to note that Liu et al. (2016) (10.1016/j.jqsrt.2015.08.005) show considerably less enhancement for the more realistic DDA models than for core-shell Mie model.

There are two issues: first, the wavelength dependence of absorption of coatings (or coated absorbing cores); second, the "loss" of the latent heat from the photo-thermal or photoacoustic signal due to the volatilization of coatings.

**Wavelength dependence**

In this study, the soot coatings consisted of secondary organic matter from the oxidation of a biogenic precursor ( $\alpha$ -pinene). According to the literature, biogenic SOA does not absorb as strongly as anthropogenic SOA. Light absorption is only observed in the UV and near-UV region:

https://www.ncbi.nlm.nih.gov/pmc/articles/PMC7372712/

https://doi.org/10.1016/S0021-8502(03)00361-6

Under certain experimental conditions, SOA from the ozonolysis of  $\alpha$ -pinene showed negligible light absorption even at near-UV wavelengths:

**https://doi.org/10.1029/2010JD014387**

Therefore, it is not surprising that the PAX at 870 nm did not show any sensitivity to coatings. We also see the increase of the AAE with increasing thickness of the coating in the PTAAM (Table 1). We interpret this as the increase in absorption at lower wavelengths (532 nm) relative to the higher wavelengths (IR). Additionally, we see with PTAAM an enhancement of absorption at 532 nm for about a factor of 1.3 (Fig. 3) and no enhancement at 1064 nm (Fig. 5).

There are some differences between different modelled results regarding this effect. Virkkula (https://doi.org/10.5194/amt-14-3707-2021) states that the lensing effect is less pronounced at longer wavelengths using a Mie model.

Liu et al. (http://dx.doi.org/10.1016/j.jqsrt.2015.08.005) show (Figure 9) that the calculated lensing effect differs between the discrete-dipole approximation (DDA) and core-shell Mie models. The DDA model predicts lower lensing at longer wavelengths; core-shell model predicts a similar magnitude of lensing from the visible to the IR.

The differences between different models can be reconciled with measurements which we present using PTAAM.

**Volatilization of coatings**

We have evaluated the volatility of our SOA coating using a total carbon analyzer, developed by FHNW, during the increase of the sample temperature (manuscript in preparation). We do not have any evidence that the SOA coatings used in our study are more volatile than typical suburban aerosols. Had the coatings we produced been volatile, this would have been seen in the comparison between the PAX and PTAAM.

PTAAM measures absorption at 532 nm and 1064 nm on the same sample (retention time for aerosol in the pump beam is 1000-times longer compared to the modulation interval). In principle, it would be possible to evaporate the coating by the pump beam in PTAAM. Increase of the measured AAE shows

that at least part of the coating has not been removed. Estimated pump beam intensity in PTAAM is 2 W/mm2 (PTAAM article in preparation – will be submitted shortly).

The recondensation of the possibly volatile coatings would occur during the off-phase of the pumping laser in the PAX and PTAAM. In the PAX, the acoustic wave, generated by absorption of light, moves away from the sample signal generation region by the time recondensation occurs. In the photo-thermal interferometer, the signal due to volatilization and recondensation are generated in the monitored probed by the probe beam, so the recondensation is measured and the latent heat is taken into account properly. We do not expect any absorption by the coating itself and no absorption enhancement by the coated absorbing cores (see arguments above). We would expect to measure the difference in the IR between the PAX and PTAAM if the SOA coatings had been volatile. We saw no differences (Fig. 5) in agreement with our total carbon measurements.

Considering other impacts of measurement uncertainty on the analysis, we provide statistical uncertainties in Figure 3 and 4 as well as the uncertainty of the splitter bias (1%, subsection 2.1). Other possible measurement uncertainties include diffusion losses in the connecting tubes. We have added the following text in subsection 2.1 (Line 110): "The PAX, PTAAM, PAS and AE33 have very similar sampling flows (1-2 L/min) and the difference in diffusion losses was compensated by adapting the length of the connecting tube to the flow of the instrument. For the MSPTI, which has a flow of 0.25 L/min, the connecting tube was kept as short as possible. Possible differences in the internal path length of the instruments (between the aerosol inlet and measurement cell) were not taken into account. In the case of the MAAP, which was operated at a flow of 12 L/min, it was challenging to compensate for the difference in the diffusion losses. We cannot rule out that the measurements by the MAAP are biased but we estimate that this bias is <5% and therefore much smaller than the systematic uncertainties of this filter-based instrument".

To conclude, we think that systematic measurement uncertainties due to splitter bias and diffusion losses have a small impact on our analysis. The discrepancy in the insitu measurements seems to be instrument-dependent. Especially the prototype PAS and MSPTI instruments suffer from artefacts arising, for instance, from instabilities in the laser power (see Lines 306-308 and 325-327 for more details). These instruments are still under development. The reason we included the MSPTI and PAS in the study was not to suggest that these instruments are "mature" enough to be employed for BC monitoring, but to show how the combination of a miniCAST with the OCU can be used to test instruments based on different technologies. Commercialised instruments, such as the PAX and PTAAM, show in most cases a reasonable agreement. Figure 3b shows that the difference between the PTAAM and the PAX in babs is 10-20%, and the two instruments report practically the same Eabs (Figure 3c).

**Reviewer:** In their analysis described on lines 313-314, the authors used the NIR wavelengths to decouple the lensing effect. If different wavelengths are used, then the size parameters differ, and the amount of lensing changes. Using the NIR wavelengths, the authors propose that absorption enhancement observed at 670 nm and shorter is due to the production of brown carbon from the oxidation of a-pinene. The authors need to back up this assertion. What species do they think is responsible for light absorption all the way out to 670 nm? The authors might want to start by looking at the paper by Song, C., et al., (2013), Light absorption by secondary organic aerosol from  $\alpha$ -pinene: Effects of oxidants, seed aerosol acidity, and relative humidity, JGR., 118, 11,741–11,749). Very few organic aerosols exhibit light absorption to such long wavelengths. Additionally, the variability among

the in situ measurements of absorption enhancement is surly large enough to warrant a discussion on how this variability impacts the conclusions drawn from the observations as discussed above.

**Response:** This is a very good comment. We believe that the absorption enhancement reported by the MAAP might be an artefact. We amended the text (Line 270) as follows: "These results suggest that the absorption enhancement observed by the AE33, MAAP, PAS, PTAAM and MSPTI could be due to light absorption by SOM. Biogenic SOM is only expected to absorb light in the UV and near UV region (Song, 2013); thus, it is surprising that the MAAP indicates absorption enhancement at 637 nm. One of the possible reasons could be coating of BC in the filter by SOM or modification of the filter matrix optical properties by SOM (Cappa et al., 2008).

Specific comments:

**Reviewer:** The authors are strongly encouraged to be very pedantic when discussing the coated soot particles. For example, the authors should explicitly distinguish in their tables uncoated soot particles and soot-containing particles (i.e., coated soot). It took a few rereads for this reviewer to fully appreciate that the reported GMDs were for the polydispersed aerosol exiting the coating/mixing chamber, irrespective of whether the soot was coated or not, and not a more direct and meaningful comparison of "uncoated" soot with "denuded" coated soot. The authors are strongly encouraged to reword some of their sentences to reflect this. For example, on page 8 (lines 230 - 231) the authors write "....The GMDmod of the soot particles increased from 88 nm to 126 nm while the EC/TC mass fraction dropped ....". A more precise communication would be "The GMDmod of the soot-containing particles increased from 88 nm to 126 nm following coating in the OCU...." I push on this because the authors point out because absent explicit distinguishing between soot and soot-containing particles and how the authors report the size distributions, someone could easily misinterpret that the reported GMDs for the coated cases as being derived from denuded soot - which is clearly not the case.

**Response:** We apologise for the confusion. In Table 1, we do distinguish between uncoated and coated particles (see column 1), but we have now added the following clarification in the text (Line 216) to avoid any misunderstandings: "Note that the uncoated soot particles are "fresh" soot particles generated by the miniCAST burner (and not soot particles that have been coated and denuded)". We have also modified Figure 1 to show that the uncoated soot particles are delivered to the BC-measuring instruments through a line that bypasses the oxidation flow reactor.

We have also modified the text (lines 220 and 238) to make it clear whether soot is uncoated or coated: "The geometric mean mobility diameter ( $GMD_{mob}$ ) of the soot particles gradually decreased from 92 nm (uncoated soot) to 83 nm (coated soot; coating 3) as shown in Fig. 2a" and "the  $GMD_{mob}$  of the soot particles increased from 88 nm (uncoated soot) to 126 nm (coated soot) while the EC/TC mass fraction dropped from ~85 % to ~10 % and the SSA increased up to ~0.7".

We agree with the Reviewer that a comparison of uncoated soot with denuded soot would be meaningful in order to better compare with atmospheric measurements. However, as the Reviewer noted in one of the comments above, denuding coated soot is not as straightforward as it sounds. Some SOA components are not volatile and might not be removed in the thermo-denuder or might be transformed by the heating, thus leaving a thin coating around the soot core. In our study, we made sure to avoid such artefacts by using fresh soot by the miniCAST burner.

**Reviewer:** It is clear that the filter-based measurements are biased quite high compared to the in situ techniques. Are the authors worried about relying on a filter-based technique to derive AAEs from which measurements at other wavelengths can be adjusted to a single wavelength?

**Response:** This is a valid concern, which was also raised by Reviewer 1. We have now revised Figs 3 and 4 of the manuscript by calculating b\_abs based on AAE values from the PTAAM. We have also revised Tables S2 and S3 accordingly.

**Reviewer:** On line 262 the authors state that R\_BC=3.4 which corresponds to EC/TC of 0.1. From the authors definitions of R\_BC, and M\_tot/M\_BC - which should be R\_BC+1 - then EC/TC =  $1/(R_BC+1)$ . R\_BC =3.4 does not give E/TC = 0.1. Please check on this.

**Response:** The values of R\_BC, and M\_tot/M\_BC are based on gravimetric measurements by TEOM as explained in the caption of Figures 3 and 4. It is true that the TEOM measurements do not agree so well with the EC/OC measurements. We believe that this is due to the high measurement uncertainties of the thermal-optical analysis and particularly with the difficulty to define the split point.

**Reviewer:** While it is likely that the BC particles are fractal-like, to get an SSA of 0.03 (for m=(2,-1)) requires a diameter of 50 nm, yet their GMD - mobility diameter - is 90 nm. The authors should explain this discrepancy.

**Response:** The (uncoated) BC particles with  $GMD_{mob} = 90$  nm are fractal-like as shown by TEM analysis (see Ess et al., 2021 main manuscript and supplemental information https://www.tandfonline.com/doi/full/10.1080/02786826.2021.1901847).

**Reviewer:** This reviewer found it interesting that under the more dilute conditions (setup 0.1) the soot-containing particles were more heavily coated - as inferred from the GMDs. Why is that? Is this simply a limitation of the coating chamber in the limit of very high seed aerosol concentrations?

**Response:** As the dilution only affects the soot concentration and not the concentration of the VOC precursor, we have generally lower soot-to-VOC ratio, resulting in thicker coatings. When we do not dilute the soot aerosols, we have a higher number concentration of soot cores acting as seed particles (and therefore a larger surface area for the SOA to condense on), leading to thinner SOA coatings.

**Reviewer:** Somewhat related to the above, given the very high concentrations (i.e., 107 cc1) of soot in the micro smog chamber during the Setup 1 experiments, are the authors worried about coagulation? Since coagulation goes as the square of number concentration, it is very likely that coagulation is competitive with condensation and the authors are encouraged to evaluate the impacts, if any, on their observations.

**Response:** We measured the size distribution of soot upstream and downstream of the micro smog chamber and found no difference when GMDmob=90 nm (even at number concentrations of  $10^{7}$  cm-3). We observe, however, coagulation when GMDmob < 50 nm and care must be taken when working with soot in that size range.

**Reviewer:** Lines 194 - 200. Introducing a filter media before discussing the measurement seems backwards, The authors are encouraged to combine and order the sentences in these two paragraphs.

**Response:** This is indeed a bit confusing. We have restructured the two paragraphs into a more logical order.

**Reviewer:** This reviewer found the legends in figures 3 and 4 a bit confusing in that the wavelength listed by the instrument is its operating wavelength, yet the data presented in the plots is adjusted to allow comparison at a single wavelength (532 nm). The authors might was to clarify this a bit better in the figures.

**Response:** We apologise for the confusion. We have amended the text in the captions of Figures 3 and 4 as follows: "The legend below Figure 3c (Figure 4c) indicates the wavelengths at which the measurements were performed".

---

## Author Response (AR2)

**EDITOR**

Thank you for your response to the initial review and the associated improvements to the manuscript. As you can see from the reviewer comments, they express some remaining concerns that should be addressed prior to publication. In particular, I find Reviewer #2's comments regarding the need to carefully quantify uncertainties and to make sure that the conclusions are robust in the face of these uncertainties compelling. I ask that you carefully consider the remaining concerns and suggestions made by both reviewers and modify the manuscript to address these issues. The revised manuscript will be sent to Reviewer #2 for further input.

This manuscript has many strong elements and represents an impressive experimental effort on a very important topic. Careful consideration of the robustness of the conclusions that can be drawn from imperfect data will make the scientific contribution only stronger. So I thank you for the efforts already made to address the reviewers' comments, and in advance for the additional changes they request.

**Response:**

**Dear Editor**

Thank you very much for your feedback. We completely agree that there is a need to quantify measurement uncertainties. The reason why we hesitated so far to provide the uncertainties of  $b_{abs}$  are the following:

 Filter-based methods, such as the aethalometer and MAAP, suffer from systematic uncertainties which are difficult to quantify. To setup a robust uncertainty budget which takes into account the wavelength of the lightsource and the particle-specific properties of the test aerosols (e.g. SSA), calibration against a reference method would be needed.

For the AE33, the GAW recommendation for part of the systematic error relative to the filter multiple scattering parameter is 25% (WMO, 2016). The cross-sensitivity to scattering, which manifests as SSA dependence, should be smaller than ~20% estimated based on our measurements and the results from (Yus-Díez et al., 2021).

2) The uncertainties of prototype instruments, such as the PAS and MSPTI, are dominated by unexpected changes of the properties of the laser irradiation (e.g. modulation depth, frequency spectrum or beam cross-section) or pump performance which are impossible to rigorously quantify.

Nevertheless, we now provide an estimation of the measurement uncertainties related to  $b_{abs}$  in the manuscript text, section 2.2 "BC- and aerosol-absorption-measuring instruments".

Please find below a point-by-point response to the Reviewers' comments.

Thank you for your time and consideration.

Lack, D. A., Cappa, C. D., Covert, D. S., Baynard, T., Massoli, P., Sierau, B., Bates, T. S., Quinn, P. K., Lovejoy, E. R. and Ravishankara, A. R.: Bias in Filter-Based Aerosol Light Absorption Measurements Due to Organic Aerosol Loading : Evidence from Ambient Measurements, Aerosol Sci. Technol., 42(12), 1033–1041, doi:10.1080/02786820802389277, 2008.

Song, C., Gyawali, M., Zaveri, R. A., Shilling, J. E. and Arnott, W. P.: Light absorption by secondary organic aerosol from  $\alpha$ -pinene: Effects of oxidants, seed aerosol acidity, and relative humidity, J. Geophys. Res. Atmos., 118(20), 11,741-11,749, doi:10.1002/jgrd.50767, 2013.

WMO: WMO/GAW Aerosol Measurement Procedures, Guidelines and Recommendations. [online] Available from: https://library.wmo.int/doc\_num.php?explnum\_id=3073, 2016.

Yus-Díez, J., Bernardoni, V., Močnik, G., Alastuey, A., Ciniglia, D., Ivančič, M., Querol, X., Perez, N., Reche, C., Rigler, M., Vecchi, R., Valentini, S. and Pandolfi, M.: Determination of the multiple-scattering correction factor and its cross-sensitivity to scattering and wavelength dependence for different AE33 Aethalometer filter tapes: A multi-instrumental approach, Atmos. Meas. Tech. Discuss., 29(March), 1– 30, doi:10.5194/amt-2021-46, 2021.

**REVIEWER 1**

Review of revised paper: "Comparing black carbon and aerosol absorption measuring instruments – a new system using lab-generated soot coated with controlled amounts of secondary organic matter"

**Manuscript**

1. Pg 1 line 24. It may be useful to give the wavelength for the SSA observation range 0 to 0.7.

**Response:** We have modified the sentence, which now reads: ... and single scattering albedo (SSA at 637 nm) from almost 0 to about 0.7".

2. Pg 6 line 145. Do the oscillations of the MAAP data correspond with filter spot changes?

**Response:** There seems to be a general consensus in the community on the source of the artifact being a non-disclosed internal averaging algorithm, but unfortunately there are no publications describing this artifact. We have amended the last sentence of the paragraph (Line 143) as follows: "While the exact reason for the variations is not known, they occur mid-range of the MAAP spot collection duration, and thus seem to be instrument-dependent and possibly related to a non-disclosed internal averaging algorithm.".

We have now highlighted the filter spot times in figure S2 to support the discussion in the manuscript and we also added the following explanation (SI, section S3): "While the oscillations coincide with the frequency of filter spot changes, the actual spot change takes place during a steadier period of the oscillations. Therefore,

as stated in the manuscript chapter 2.2, the oscillations are not related e.g. to the spot-change related artefact described by (Hyvärinen et al. 2013)".

3. Pg 16 lines 368-369. One issue with using the low-cost air quality sensor as a nephelometer is that it measures at close to ambient relative humidity (unless temperature controlled) so that it may not represent aerosol scattering that affects the AE33.

**Response:** Thank you for pointing this out. We believe that the relative humidity needs to be addressed explicitly as well. We have amended the text (Line 365) as follows: "By letting a low-cost nephelometer (temperature- and RH-controlled) run parallel to the AE33 at monitoring stations, ...".

4. Pg 17 line 376. It may be useful to give the wavelength for the SSA range quoted.

**Response:** We have amended the text (Line 373) as follows: "... and optical properties (SSA almost 0 to 0.7 at 637 nm). ".

**Supplemental Information**

5. First sentence of section S4: There is a reference to an acoustic resonator in Figure S3, though the figure does not show a resonator.

**Response:** We have merged the pictures S3 and S4 and added a real picture of the setup. The resonator is now visible.

6. Line 45, 'photoacoustic' should be 'photoacoustic signal'.

**Response:** Thank you for spotting this. The sentence now reads: "The photoacoustic instrument uses a novel resonator chamber with elliptical cross-section to enhance the photoacoustic signal, ...".

7. Line 46. How were the 3 lasers combined?

**Response:** We have added the following explanantion (Line 61, SI): "The laser wavelength is switched periodically every 60 seconds from blue to green to red, using only one wavelength at a given time. The three laser beam paths were combined inside the laser housing by way of dichroic mirrors".

8. Line 49. I'm not sure what is meant by the "attenuation position". Do you mean 'excitiation position."?

**Response:** Yes, we mean excitation. We have corrected the text accordingly.

9. In Figure S4, how are the aerosol put into the resonator? Is the ellipse a 1 D tube with elliptical cross section? What is the response time of the instrument to sudden aerosol inputs? What are the dimensions of the resonator?

**Response:** The new Figure S3 should be clearer now. We have also added: "Aerosols enter at one end of the resonator, and are drawn out at the other end using a pump. Response times to sudden aerosol inputs were in the order of three to four minutes." Regarding the dimensions, we added the following explanation: 10 cm width, 8 cm height (see Line 58, SI).

10. Line 59. "...attenuates the acoustic modes ...". Do you mean "... excites the acoustic modes ..."?

**Response:** Yes, we apologise for this mistake. We have corrected the text (Line 77) accordingly.

11. Line 59. What kind of loudspeaker is used to generate sound at 22.7 kHz? How is it coupled to the chamber to excite the relevant mode?

**Response:** We now specify that it was a Balanced Armature Driver WBFK-30095-000 (Line 76), and added the following sentence (Line 67): "The loudspeaker and the microphone are both guided with the help of a rod into the resonator chamber."

12. Line 65-70. Since the phase delay was not measured, how is the aerosol light absorption made quantitative with this instrument?

**Response:** The PAS is calibrated with 1 ppm NO2 and the aerosol signal amplitude is compared to the signal amplitude from the NO2 measurements.

The PAS is a prototype instrument, which is still in the development phase. Further efforts are needed to improve the stability and repeatability of the measurements. Uncertainties are currently dominated by sporadic technical errors, which led to incomplete measurements for some data points.

We have now removed the PAS data from Figs. 3-4 of the main manuscript. The data are still listed in Tables S2 and S3 in the Supplemental Information.

\_\_\_\_\_

**REVIEWER 2**

The revisions undertaken by the authors have strengthen their manuscript. However, this said, this reviewer still has an issue with the "definitive tone" taken by the authors with respect to attributing light absorption enhancement at the shorter wavelengths to coating absorption based on the absence of light absorption at the NIR wavelengths.

While DDA does predict a weaker light absorption enhancement relative to the simplistic core-shell model, the light absorption enhancement due to transparent coatings does not go to zero at the NIR wavelengths. Given the over all measurement uncertainty (which must include uncertainties associated with the light absorption measurements themselves, measurements of the aerosol mass concentrations and the derived mass ratios of coating to core, etc.) the conclusive tone that the light absorption enhancement observed at the shorter wavelengths is

due to coating absorption is not justified. The authors are strongly encouraged to review the work of Lack and Cappa (ACP, 10, 4207-4220, 2010) wherein those authors reported that "....BC cores coated in CClear can reasonably have AAE of up to 1.6, a result that complicates the attribution of observed light absorption to CBrown within ambient particles. However, an AAE < 1.6 does not exclude the possibility of CBrown; rather CBrown cannot be confidently assigned unless AAE > 1.6." (Lack and Cappa abstract) The reported AAEs contained in Table 1 range from 1.14 - 1.48 for the Aethalometer-derived values and from 0.875 - 1.36 for PTAAM-derived values, both of which fall well below the canonical 1.6 threshold cited by Lack and Cappa. This reviewer is not calling into the question the results, but rather the tone of certainty that the authors take with respect to either discussions of light absorption enhancement or attribution. Examples include: "As expected..." (line 270); "...the PAX which is insensitive to coating..." at 870 nm (line 263); or "These results suggest that the absorption enhancement....is dominated by light absorption by SOM" (lines 274-275). As alluded to above, the derived AAEs do not warrant such a conclusive tone. Additionally, the authors seem to have glossed over this reviewer's comment regarding the fact that the size parameter will differ for differing wavelengths and thus caution should be exercised when extrapolating the absence of light absorption at the longest wavelengths as evidence that light absorption observed at the shorter wavelengths IS due to light absorption by the coating. Indeed, this issue was also raised by Reviewer 1. The attribution issue needs to be addressed for fully before this manuscript can deemed as acceptable for publication. This reviewer believes that the work of Lack and Cappa along with the size parameter dependence on light absorption are likely the more robust explanations for the observed light absorption at shorter wavelengths versus coating absorption - based on the data presented in the manuscript.

**Response:** We thank the Reviewer once more for the valuable feedback. We apologise if the tone sounded too definitive, it was not in our intentions. Our intention was not to claim that the absorbing coatings would result in absorption enhancement and the text around line 270 was an oversight. We explicitly mention non-absorbing coatings in the replies to Reviewers and mention this only as a possible (general) mechanism in the manuscript text. We completely agree with the Reviewer and have modified the text and used a more reliable measurement of the AAE.

Indeed, now that we use the AAE values from the PTAAM (instead of the AE33) to convert  $b_{abs}$  of the PAX to 532 nm, the PAX also reports a weak absorption enhancement. We have now deleted the sentence: "...the PAX which is insensitive to coating..." at 870 nm (line 263) and have modified the text of the paragraphs as follows:

"In the visible and near UV region of the spectrum, the values of  $E_{abs}$  can include effects of both "lensing" and potential absorption by SOM. Absorption by  $\alpha$ -pinene-derived SOM is very low with a MAC below 0.25 m2g-1 (Nakayama et al., 2010) or even 0.01 m2g-1 (Lambe et al., 2013) at 532 nm, depending on the oxidation state and experimental details. Instruments measuring in the wavelength region 520–637 nm all recorded an increase in  $E_{abs, 532}$  as a function of  $R_{BC}$  (Fig. 3c). At  $R_{BC} \approx 3.4$ , corresponding to an EC/TC mass fraction of 10 % and an SSA of about 0.7, an absorption enhancement in the range 1.3 (PTAAM 532 nm) to  $\sim$  2 (MSPTI 532 nm) was observed.

A weak absorption enhancement of about 1.1-1.3 at 532 nm was calculated from the PAX data (Figure 3c). We therefore interpret the absorption enhancement shown in Figure 3c to be due to a transparent coating by SOM on the absorbing BC core, as described by Lack and Cappa (2010). Moreover, as biogenic SOM is only expected to absorb light in the UV and near UV region (Nakayama et al., 2010, Lambe et al., 2013, Song et al., 2013), it is surprising that the MAAP indicates such a pronounced absorption enhancement at 637 nm. Apart from the lensing effect, one additional reason could be coating of BC in the filter by SOM or modification of the filter matrix optical properties by SOM (Lack et al., 2008)."

To return to the absorption of coatings, we would like to make a general observation that "browness" of BrC should not be interpreted as the absorption of the coating at the lower wavelengths, but rather the coated particles absorbing in this region – with the naming of BrC based on the properties of the whole particle, which is a coated soot core.

Concerning the measurement uncertainties:

- 3) The statistical uncertainty in  $R_{BC}$  was listed in Table S2 (as standard deviation). Here, we have made the following correction: we now provide the uncertainty as standard deviation of the mean in order to be in line with the rest of the manuscript. We also provide an estimation of the combined measurement uncertainties of the TEOM and  $R_{BC}$ , respectively:
- 4) (Line 197) TEOM measurements agreed within 1%-4% with the reference (manual) gravimetric method.
- 5) (Captions of Figs 3 and 4)The uncertainty (*k*=1) in *R*BC is estimated to be about 5% (not shown).
- 6) We now provide an estimation of the measurement uncertainties related to babs in the manuscript, section 2.2 "BC- and aerosol-absorption-measuring instruments". We have revised Figures 3-5 accordingly. Note that the data points in panels b) and c) have been slightly shifted along the x-axis to improve the readability of the graph (the correct *R*BC values are listed in Tables S2-S4). A clarification has been added in the caption of the figures.

We come to the following conclusion:

(Line 262): Even when taking into account the expanded measurement uncertainties (k=2; 95% confidence interval), the measurements by the AE33 hardly agree with the measurements by the PAX and PTAAM. This indicates that the ~20% measurement uncertainty (k=1) assigned to the AE33 (see section 2.2) might be underestimated. Similar observations can be made for the MAAP at high  $R_{BC}$  ratios even though the deviations from the PAX and PTAAM are less pronounced.

We have also amended the text as follows:

(Line 277): The uncertainties in Figure 3c were calculated as the quadratic sum of the uncertainties in  $b_{abs}$  for the uncoated and coated soot. Note that this procedure is only a simplistic approximation. Ideally, the uncertainty in  $b_{abs}$  should be partitioned in type A (random) and type B (systematic) uncertainties and correlations between the different components should be taken into account. A robust uncertainty calculation was, however, not possible because the uncertainties of the instruments are not so clearly understood and, additionally, instruments such as the PAS and the MSPTI at times suffered from unexpected technical errors. In the case that  $b_{abs}$  is dominated by systematic uncertainties which remain the same when measuring the uncoated and coated soot particles, such uncertainties may cancel out, resulting in a much smaller combined uncertainty in  $E_{babs}$  than what presented in Figure 3c.

**References:**

- Nakayama, T., Y. Matsumi, K. Sato, T. Imamura, A. Yamazaki, and A. Uchiyama (2010), Laboratory studies on optical properties of secondary organic aerosols generated during the photooxidation of toluene and the ozonolysis of a-pinene, J. Geophys. Res., 115, D24204, doi:10.1029/2010JD014387.
- 2) Andrew T. Lambe, Christopher D. Cappa, Paola Massoli, Timothy B. Onasch, Sara D. Forestieri, Alexander T. Martin, Molly J. Cummings, David R. Croasdale, William H. Brune, Douglas R. Worsnop, and Paul Davidovits: Relationship between Oxidation Level and Optical Properties of Secondary Organic Aerosol, Environ. Sci. Technol. 2013, 47, 6349–6357, dx.doi.org/10.1021/es401043j, 2013.
- Lack, D. A. and Cappa, C. D.: Impact of brown and clear carbon on light absorption enhancement, single scatter albedo and absorption wavelength dependence of black carbon, Atmos. Chem. Phys., 10, 4207–4220, https://doi.org/10.5194/acp-10-4207-2010, 2010

**Specific comments:**

Line 150. The authors write: "....the FHNW group uses three different wavelengths (445 nm, 520 nm, 638 nm, ~300 mW each)...." Yet in the supplemental the authors cite - on line 47 - that the power level for the 520 nm is 700 mW, no where close to 300 mW. Which is it?

**Response:** Apologies for this typo, we meant 300 mW and we have corrected the text accordingly (SI, Line 60).

Line 218. The potential impacts of  $10^{7}$  /cc concentrations leading to coagulation. The rate of coagulation is proportional to the square of the number concentration. It is hard to imagine that coagulation is not occurring at such high concentrations, especially when coagulation has been observed in other studies at lower concentrations (~10^5 /cc). Perhaps the absence is due to transit time in the coating chamber? The authors are encouraged to at least speak to the possibility of coagulation and why they think it is not present.

**Response:** In Table S1 two different operation points for the miniCAST are listed. For Setup 1 (no diluter between the miniCAST and oxidation flow reactor), we had to slightly modify the settings of the miniCAST in order to still generate soot with GMDmob of 90 nm. Without modifying the setting, we would obtain particles with GMDmob > 90 nm due to coagulation. We believe,

however, that coagulation happens already in the outlet pipe of the miniCAST (and perhaps in the tube connecting the miniCAST with OFR). No further coagulation was observed in the OFR most probably because of the short residence time of the aerosols in the quartz tube (about 3 s). We have added an explanation in Section S1.

Line 268. This is where authors state that R\_BC=3.4 which corresponds to EC/TC of 0.1. In their response the authors state that "It is true that the TEOM measurements do not agree so well with the EC/OC measurements. We believe that this is due to the high measurement uncertainties of the thermal-optical analysis and particularly with the difficulty to define the split point" The authors are encourage to put the response into the manuscript, because an interested reader with will the same simple calculation and discover ~ 2x difference between the MBC/MTotal derived form  $r_BC=3.4$  and the reported EC/TC = 0.1.

**Response:** We have added this clarification in Line 247.

Line 344: The authors cite the modest enhancement in light absorption of ambient aerosols reported by Nakayama who conducted measurements at 781 nm. While this review is NOT a review of the Nakayama work, caution must always be exercised when comparing light absorption measurements using a denuder as denuders are known not to remove all the coating, yet making the assumption that the coating is all vaporized.

**Response:** The reason why we compared with the work of Nakayama et al. is because the authors studied aerosols with thin coatings. In our work, we also generated thin to moderate SOA coatings. But we agree with the Reviewer that it is quite difficult to compare laboratory with field studies because of the different conditions under which the aerosols are generated and processed. We have now removed the reference to Nakayama et al. from our manuscript.